# h1: Bootstrapping LLMs to Reason over Longer Horizons via Reinforcement Learning

**Alesia Ivanova** [* 1]  **Sumeet Ramesh Motwani** [* 1]  **Ziyang Cai** [2]  **Philip Torr** [1]
**Riashat Islam** [3]  **Shital Shah** [3]  **Christian Schroeder de Witt** [† 1]  **Charles London** [† 1]

Correspondence: `sumeet.motwani@eng.ox.ac.uk`, `charles.london@cs.ox.ac.uk`

## Abstract

Large language models excel at short-horizon reasoning tasks, but performance drops as reasoning horizon lengths increase. In this work, we introduce a scalable method to bootstrap long-horizon reasoning capabilities using only existing, abundant short-horizon data. Our approach synthetically composes simple problems into complex, multi-step dependency chains of arbitrary length. We train models on this data using outcome-only rewards under a curriculum that automatically increases in complexity, allowing RL training to be scaled much further without saturating. Empirically, our method generalizes remarkably well: curriculum training on composed 6th-grade level math problems improves accuracy on longer, competition-level benchmarks. It also transfers significantly to diverse out-of-distribution ReasoningGym domains and long-context benchmarks, indicating broader generalization. Importantly, our long-horizon improvements are significantly higher than baselines *even* at high *pass@k*, showing that models can learn new reasoning paths under RL. Theoretically, we show that curriculum RL with outcome rewards could achieve an exponential improvement in sample complexity over full-horizon training, providing training signal comparable to dense supervision. *h1* therefore introduces an efficient path towards scaling RL for long-horizon problems using only existing data.

## 1. Introduction

Large language models (LLMs) have improved remarkably across many domains, but they often struggle with long-horizon reasoning (LHR). This involves correctly carrying out a deep multi-step reasoning process where goals must be decomposed into intermediate steps and executed successfully in a chain of thought (CoT). LHR tasks consist of a sequence of dependent problems where a large context must be managed and errors can compound across the horizon (Li et al., 2024; Malek et al., 2025; Zhou et al., 2025a; Sinha et al., 2025). For many problems of interest, such as performing research-level mathematics, debugging complex code, and assisting with scientific discovery, an LLM must be able to correctly solve intermediate steps, carry forward results, and determine what state is important to track and use (Lu et al., 2024; Zhang et al., 2025c). Broadly, any task of importance requires solving several challenging interdependent components, which motivates the development of training methods directly aimed at improving model capabilities on such long sequences of problems.

Reinforcement learning (RL) has shown substantial benefits in improving the reasoning capabilities of LLMs (OpenAI-o1 et al., 2025; DeepSeek-AI et al., 2025). However, RL depends on the availability of verifiable training data and is currently limited in terms of the complexity and long-horizon reasoning paths afforded by such data. The lack of increasing levels of problem difficulty and diversity in RL datasets leads to improvements saturating after a very limited number of training steps (Cui et al., 2025; Wu et al., 2025). Obtaining long-horizon training data is expensive and sample inefficient to directly train on (as we show in Section 4 and Appendix B). Moreover, existing approaches (Zhang et al., 2025a; Liu et al., 2025b) do not adequately address the problem of improving long-horizon performance when only single step short-horizon data is abundant (as is the case in real-world scenarios). This raises a natural question: ***Can we improve long-horizon reasoning capabilities by scaling reinforcement learning using only existing short-horizon training data?***

---

*Equal contribution †Equal advising. [1]University of Oxford [2]Princeton University [3]Microsoft AI Frontiers. Correspondence to: Sumeet Ramesh Motwani <sumeet.motwani@eng.ox.ac.uk>, Charles London <charles.london@cs.ox.ac.uk>.

*Proceedings of the 43rd International Conference on Machine Learning*, Seoul, South Korea. PMLR 306, 2026. Copyright 2026 by the author(s).

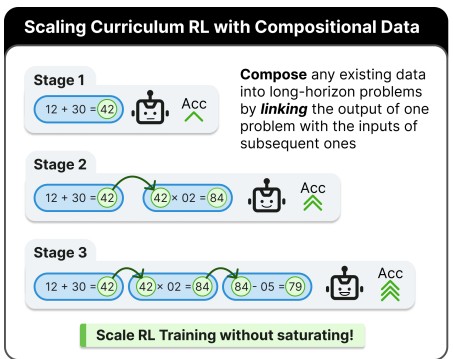 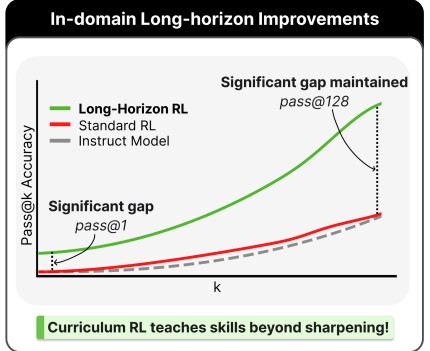 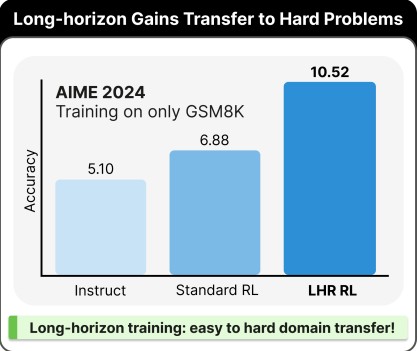

*Figure 1.* Our method $h1$ is simple and involves composing existing short-horizon reasoning problems into longer sequences. We then apply a stage-wise curriculum on this composed data to scale online RL training. We observe significantly improved long-horizon reasoning capabilities and OOD improvements on harder math benchmarks, reasoning benchmarks, and long-context benchmarks.

In this work, we show that the answer is *yes*. We introduce a method for chained problem construction, which composes short-horizon problems (e.g. GSM8K questions (Cobbe et al., 2021)) into arbitrarily long chains of dependent reasoning steps. This provides scalable synthetic long-horizon data, with explicit control over the horizon length and complexity without the need for new annotations. We then train language models on this data using reinforcement learning with outcome-only rewards, coupled with a curriculum over horizons. Obtaining useful data that is of just the right complexity for models to learn from has always been a major bottleneck (Li et al., 2025). We show how existing tasks can be grouped adaptively into increasingly harder problems that provide useful training signal and prevent RL improvements from quickly saturating (Cui et al., 2025). Our approach requires neither step-level labels nor PRMs, and avoids inference-time search, instead directly training the model to internalize useful long-horizon reasoning paths.

Our results in Sections 4 and 5 show that not only does this synthetic curriculum generalize to long in-domain multi-step problems, but it also transfers to harder benchmarks with longer problems such as MATH-500 and AIME that implicitly require LHR. We also observe significant generalization on diverse out-of-distribution ReasoningGym domains (Stojanovski et al., 2025) and long-context benchmarks. Importantly, we show that long-horizon reasoning depends on more than just improving single step accuracy, and in Section 3 we present our method that directly targets the capabilities needed for better generalizable performance. We evaluate our long-horizon trained models versus other strong baselines *up to pass@128* and show that while improvements obtained from RLVR on standard data are bounded by the base model's capabilities (Yue et al., 2025), our method always performs significantly better. These improvements reflect new skills acquired through curriculum-based training on compositional tasks, supported by both empirical evidence and a detailed theoretical analysis (Appendix B). Finally, in Section 6, we further analyze

compute–data tradeoffs, showing how scaling compute can substitute for scarce long-horizon data in real-world settings.

Our main contributions are:

1. A general method for constructing long-horizon reasoning data by chaining existing short problems with no additional human or teacher-model annotations.

2. A reinforcement learning framework with curriculum learning and outcome-only rewards that significantly improves long-horizon performance and teaches new reasoning paths not elicited otherwise, at *pass@1* and even very high *pass@k*.

3. Empirical evidence of transfer to significantly harder out-of-distribution benchmarks (MATH-500, AIME, GSM Symbolic, ReasoningGym, LongBench-v2, Hash-hop) while training on composed *GSM8K* data, outperforming other RL training baselines.

4. Theoretical analysis of the sample complexity of curriculum learning, demonstrating that it could achieve an exponential improvement over full-horizon training, similar to dense rewards.

## 2. Related Work

**LLM Reasoning and RL.** Initial reasoning literature (Zelikman et al., 2022) bootstrapped performance using model generated reasoning traces. More recently, (OpenAI-o1 et al., 2025; DeepSeek-AI et al., 2025) demonstrated substantial improvements in reasoning capabilities via RL training. These advances have enabled effective scaling of inference-time compute (Snell et al., 2024; Brown et al., 2024; Muennighoff et al., 2025). However, as reasoning chains grow longer, models exhibit several limitations, often struggling with simple multi-step problems (Malek et al., 2025; Shojaee et al., 2025; Song et al., 2025). Moreover, RL-based approaches face their own challenges: diversity

degradation during training (Song et al., 2025), questions about whether models truly acquire new capabilities versus better sampling existing ones (Yue et al., 2025), and maintaining stability over long horizons (Xiang et al., 2025). Recent efforts toward addressing these challenges include Setlur et al. (2025), which improves in-context exploration via an RL curricula with steps such as verification and refinement and work on adaptive difficulty scheduling (Shi et al., 2025; Wang et al., 2025b; Parashar et al., 2025; Liu et al., 2025b). Our work systematically composes existing short-horizon problems into chains of increasing length, producing new data to scale an RL curriculum to train models to internalize LHR capabilities that they otherwise lack. This enables reliable multi-step problem solving and improvements on significantly harder (unseen) settings, providing a foundation for training long-horizon agents (Zhou et al., 2025b; Kwa et al., 2025). Yuan et al. (2025) appears concurrently and also looks into new skill learning during RL with compositional data.

**Length Generalization.** Length generalization is concerned with extrapolating to longer sequence lengths than those seen during training (Dubois et al., 2019; Hupkes et al., 2020; Newman et al., 2020; Anil et al., 2022). Length generalization settings mostly focus on small scale tasks (Sabbaghi et al., 2024; Zhou et al., 2024) but do not address RL training of reasoning models. A close example (Lee et al., 2025) uses curriculum construction and SFT to train small transformers on progressively harder algorithmic tasks. In this work, we not only show progressive length generalization gains through curriculum based RL, but also cross-task generalization on much harder tasks.

**Long Context Models.** Another related thread is extending LLM context length to handle very large inputs. Recent models feature context windows of tens or hundreds of thousands of tokens (Liu et al., 2025a) and benchmarks like LongBench-v2 (Bai et al., 2025) evaluate performance on extremely long inputs such as documents and code. Frontier models with state-of-the-art context windows still suffer performance degradation when required to infer against distant pieces of information or a series of dependent tasks (Li et al., 2024; Malek et al., 2025; Zhou et al., 2025a). These works show that simply having larger context windows does not guarantee that models can perform deep, dependent reasoning over several steps. Our work aims to address this gap by focusing on training for improved long-horizon output generation rather than just long-context input handling.

## 3. Method

Long-horizon reasoning (LHR) refers to the capability of reliably carrying out a stateful reasoning process over multiple dependent steps in a CoT to solve long-horizon tasks.

**What counts as a long-horizon task?** We use two notions. *Explicit-horizon* tasks have a known number of dependent sub-problems $h$ and we construct them by chaining atomic problems (used for training and in-domain evaluation). *Implicit-horizon* tasks require multiple dependent reasoning steps but do not come with an explicit decomposition (e.g., MATH-500, AIME); they have a latent horizon $h^\star$ that is not annotated. Our training targets explicit horizons for controllable generation, and shows a strong transfer to implicit-horizon benchmarks. Consistent with Sinha et al. (2025), single-turn tasks can be considered long-horizon if they consist of a large number of interdependent and difficult problems (with many dependent state updates) to solve over a long output. Operationally, a task can be considered *long* for a model when the number of dependent state updates (steps) exceeds what it can accurately solve in its CoT or compress into parameters, and our training method expands this range. This differs from generic "multi-step" settings where steps may be shallow, order-agnostic, or independent.

Our goal is to *bootstrap* long-horizon reasoning (LHR) using only existing short-horizon data. We (i) compose atomic problems into longer chains of problems with dependent steps to synthesize LHR data, (ii) scale RL training with outcome-only GRPO following a curriclum learning approach, and (iii) evaluate both in-domain (explicit chains) and on harder out-of-domain tasks that implicitly require many reasoning steps. Here, we describe what we mean by long-horizon reasoning, formalize our data construction process, and provide a description of our RL training objective.

**Atomic tasks and serial composition.** We begin with *atomic tasks* $f_j$: short, self-contained problems (e.g., single GSM8K questions) with verifiable answers that the base model solves with non-trivial accuracy. Each task takes an input $x_j$ and produces an answer $y_j$.

To form long-horizon examples, we chain $h$ atomic tasks so later sub-problems depend on earlier results. A lightweight *adapter* $\phi_j$ maps $y_j$ to the next input:

$$y_j = f_j(x_j), \qquad x_{j+1} = \phi_j(y_j), \qquad j = 1, \dots, h-1,$$

yielding the final answer $y_h = f_h(\phi_h(\cdots \phi_1(f_1(x_1))))$. Adapters may be identity or simple deterministic transforms such as scaling (addition/subtraction) or unit conversion (described further in Appendix A). Each chain of length $h$ is rendered as a *single* prompt listing the $h$ sub-problems in order. This ends up as one sequence of composed sub-problems where each sub-problem's input depends on solutions to the previous sub-problems. Correctly solving the entire problem therefore requires solving each sub-problem correctly while managing state, which can be supervised using only the final answer $y_h$ (outcome-only RL). We apply basic well-posedness checks (type consistency, unit compatibility, de-duplication) and present an example.

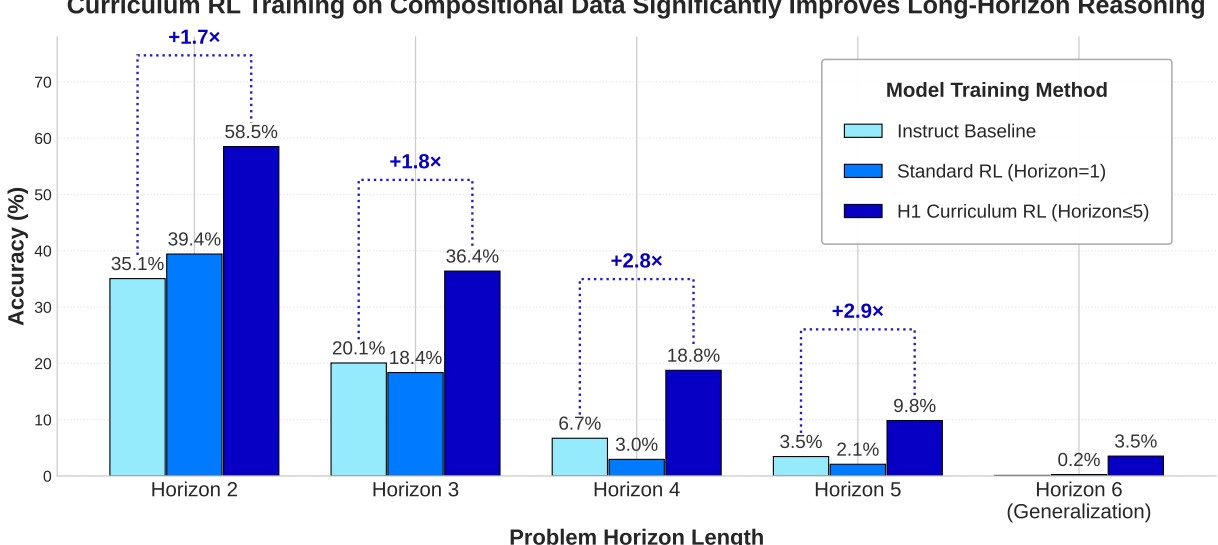

*Figure 2.* Curriculum RL training on compositional data offers significant in-domain long horizon reasoning gains (**up to 2.9×**). This prevents RL training from saturating and uses no new data.

---

**Example explicit-horizon problem**

**Problem:** Compute the final value through the following chain:

**(i)** Weng earns \$12 an hour for babysitting. Yesterday, she babysat for 50 minutes. How much did she earn? (**Answer #1**)

**(ii)** Betty is saving money for a new wallet which costs {10 × **Answer #1**} but has only half of the money needed. Her parents give her \$15, and her grandparents give her twice as much. How much more money does Betty need to buy the wallet? (**Answer #2**)

**(iii)** James writes a {**Answer #2**}-page letter to 2 different friends twice a week. How many pages does he write a year? (**Answer #3**)

**Final answer: Answer #3**

---

This construction exposes models to dependency chains that require carrying, transforming, and reusing intermediate values, while keeping supervision outcome-only. We vary chain length $h$ to implement the stagewise curriculum described later in this section. In Appendix A, we analyze our composition method through computational graphs to explain its effectiveness during training.

**Why horizons are hard: beyond multiplicative errors.** In *explicit-horizon* tasks, let $h$ be the number of dependent sub-problems whose intermediate values are reused downstream. An independent-errors view gives

$P(\text{final correct}) = p^h$, suggesting that raising atomic step accuracy $p$ suffices. We believe this is overly optimistic, and instead hypothesize that there exist specific *horizon-dependent* skills (such as intermediate value tracking and state management) that cannot be learned solely by training on atomic problems. We therefore model long-horizon accuracy via *atomic reliability* $p$ and *horizon-dependent reliability* $\sigma_j$. This matches what we see in Section 4. Abstractly, $\sigma_j$ represents skills that are necessary to achieve high accuracy at horizon $j$, beyond what is needed at horizons $< j$. Writing $s_j$ for the probability that the reasoning state remains correct after step $j$, we would have

$$s_j = p\,\sigma_j\,s_{j-1}, \qquad s_0 = 1,$$

so if $\sigma_j$ decays with horizon length, accuracy can collapse even when $p \approx 1$. This explains the weakness of naive outcome-only training at horizon $h$: when $\sigma_j \ll 1$, few rollouts earn reward, gradients have low signal-to-noise ratio, and samples scale exponentially in $h$. Curriculum training mitigates this by starting with short chains where $s_j$ is large, yielding high-SNR updates; early stages raise $p$, while later stages improve their respective $\sigma_j$. Empirically (Section 4), we validate that performance depends on capabilities beyond $p$, and our approach improves both $p$ and $\sigma_j$, delivering large gains on *explicit-horizon* tasks and generalizing to harder *implicit-horizon* tasks (Section 5); Appendix B develops the theory showing the exponential to polynomial time speedup offered by RL curriculum training and comparing it with dense supervision.

**Scaling RL with a curriculum over horizons.** Let $\mathcal{D}_h$ be the dataset of synthesized chains of explicit horizon

*Table 1.* GSM8K accuracy by horizon length. Curriculum-based RL training **significantly improves** in-domain performance compared to the Instruct model, all other equal compute baselines, and a Tree-of-Thought (Yao et al., 2023) baseline using the Instruct model.

| Model / setting | L-1 | L-2 | L-3 | L-4 | L-5 | L-6 | L-7 | L-8 |
|---|---|---|---|---|---|---|---|---|
| | \multicolumn{8}{c}{**Accuracy on GSM8K Problems of Horizon L-$n$**} | | | | | | | |
| **Instruct model** | 82.79 | 35.06 | 20.07 | 6.70 | 3.57 | 0.00 | 0.79 | 0.00 |
| *Equal compute training baselines* | | | | | | | | |
| **Only-L1** | 86.80 | 37.14 | 21.43 | 6.70 | 3.87 | 0.25 | 0.00 | 0.00 |
| **Uniform-Mix** | 82.80 | 12.66 | 2.04 | 0.54 | 0.00 | 0.00 | 0.00 | 0.00 |
| **Only-Long** | 82.71 | 43.36 | 20.41 | 3.22 | 1.49 | 0.25 | 0.25 | 0.00 |
| *Increased Inference Compute Baseline* | | | | | | | | |
| **Tree of Thought** | 83.30 | 39.40 | 13.30 | 2.00 | 0.00 | 0.00 | 0.00 | 0.00 |
| *Curriculum training (trained up to Len-$n$)* | | | | | | | | |
| **RLVR** | 83.24 | 39.42 | 18.37 | 2.95 | 2.08 | 0.25 | 0.79 | 0.00 |
| **Len-2** | 85.92 | 56.22 | 28.57 | 12.06 | 6.25 | 1.26 | 0.79 | 0.49 |
| **Len-3** | 84.91 | 56.22 | 37.76 | 15.55 | 8.63 | 3.27 | 3.17 | 0.25 |
| **Len-4** | 85.48 | 57.05 | **40.14** | 18.23 | 9.23 | 3.53 | 3.17 | 1.72 |
| **Len-5 (h1)** | **85.97** | **58.51** | 36.39 | **18.77** | **9.82** | **3.53** | **3.17** | **2.22** |
| | (+3.8%) | (+66.9%) | (+81.3%) | (+180.1%) | (+175.1%) | (++) | (+301.3%) | (++) |

$h$. Training proceeds according to a stagewise curriculum. For each horizon length $h \in \{1, 2, \ldots, H_{\max}\}$, we run Dr. GRPO (Liu et al., 2025c) on $\mathcal{D}_h$ for $s_h$ optimization steps. We initialize from $\pi_{\theta_0}$ and carry the parameters forward between stages. Algorithm 1 describes the training process.

---

**Algorithm 1** $h1$: Stagewise curriculum RL over explicit-horizons

---

**Require:** Pretrained model $M_0$; atomic task bank $\mathcal{A}$; adapters $\{\phi_j\}$; max horizon $H$; per-stage sample,step counts $n_h, s_h$
1: **for** $h = 1$ to $H$ **do**  ▷ stagewise curriculum over explicit horizons
2:    $\mathcal{D}_h \leftarrow \emptyset$
3:    **for** $i = 1$ to $n_h$ **do**  ▷ construct horizon-$h$ chains
4:       sample $(f_{1:h}, x_1)$ from $\mathcal{A}$;    $y_1 \leftarrow f_1(x_1)$
5:       **for** $j = 1$ to $h - 1$ **do**
6:          $x_{j+1} \leftarrow \phi_j(y_j)$;  $y_{j+1} \leftarrow f_{j+1}(x_{j+1})$
7:       $p \leftarrow \text{FORMATPROMPT}((f_j, x_j)_{j=1}^h)$  ▷ format prompt from the task sequence
8:       $\mathcal{D}_h \leftarrow \mathcal{D}_h \cup \{(p, y_h)\}$
9:    $M_h \leftarrow \text{TRAINWITHDRGRPO}(M_{h-1}, \mathcal{D}_h, s_h)$
10: **return** $M_H$

---

By focusing optimization on a single horizon per stage, the model first acquires reliable short-horizon primitives (increasing $p$), then learns to reuse and compose them under longer dependencies (increasing $\sigma_j$ for $j > 1$). We contrast curriculum with three baseline horizon-sampling policies:

1. **Only-L1:** $q(\ell) = \mathbb{I}[\ell = 1]$. If direct problem-solving were sufficient, this would match curriculum; empirically it does not.

2. **Uniform-Mix:** $q(\ell) \propto \mathbb{I}[1 \leq \ell \leq H_{\max}]$, i.e., randomly pick from the LHR dataset.

3. **Only-Long:** $q(\ell) = \mathbb{I}[\ell = H_{\max}]$, i.e., train solely on the hardest chains. This suffers from extreme sparsity and unstable gradients.

Generally, RL with verifiable rewards (RLVR) requires a clean labeled dataset. What models can learn from is potentially limited by the complexity expressed in these problems. We see this bound due to a fixed RL dataset both empirically (Section 3) and theoretically (Appendix B), which leads to performance quickly saturating during training. Our goal with a synthetic curriculum is to optimally utilize limited existing data for scaling RL. At each stage, tasks can be composed to be right at the edge of what a model can solve, making RLVR more scalable (see Tables 1 and 2).

**Training and evaluations.** We use the Qwen-2.5-3B Instruct model (Qwen et al., 2025) for our core experiments. Improving an Instruct model with RL is generally considered more difficult (Wang et al., 2025a) and gains signify performance improvements beyond just instruction tuning (which cannot be directly inferred for improvements on base models (Shao et al., 2025)). Therefore, we report all improvements on Instruct models for robustness. Our *explicit-horizon* training and evaluations use composed GSM8K questions (Cobbe et al., 2021), and our *implicit-horizon* evaluations are on AIME 2024, AIME 2025, MMLU Pro Math (Wang et al., 2024), GSM Symbolic (Mirzadeh et al., 2025), and MATH-500 (Hendrycks et al., 2021).

## 4. In-domain results and the importance of curriculum

We evaluate our curriculum-based RL training method using explicit-horizon GSM8K problems and demonstrate that **(i)** curriculum learning is essential for long-horizon reasoning, **(ii)** LHR performance depends on capabilities beyond single step accuracy, and **(iii)** our method teaches genuinely new capabilities that are otherwise absent in the model. We use **Qwen-2.5-3B Instruct** for our experiments, with GRPO over a curriculum of chained GSM8K problems with hori-

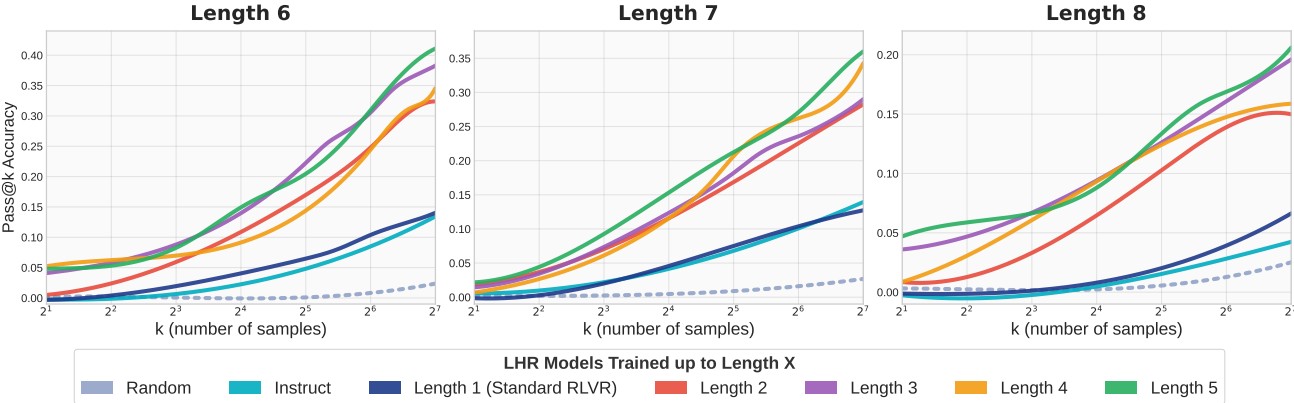

*Figure 3.* Our curriculum based RL training using composed synthetic data outperforms RLVR on standard data from the same set even at *pass@128*, **teaching new capabilities previously not in the base model**. LHR requires going beyond improving single-step performance.

zons $h \in \{1, 2, 3, 4, 5\}$. Each stage trains for 200 steps with 200 samples per horizon. We compute the following baselines: **Only-L1** (standard RL on $h$=1), **Only-Long** ($h$=5), and **Uniform-Mix** (uniform over $h \in [1, 5]$). Compute matched baselines are trained using up to the same number of training tokens as our method, and the best checkpoints are chosen by validation accuracy. The data comes from the same training distribution (where Only-L1 refers to training on horizon-1 problems, Only-Long on composed horizon-5 problems, and Uniform-Mix to training on the same problems as our method but uniformly shuffled without a curriculum). In Appendix D, we provide results on *Qwen-2.5-7B Instruct* using composed MATH and *Llama-3.2-3B Instruct* using composed GSM8K data, both showing improvements.

**In-domain results.** In Table 1, our in-domain results on composed LHR GSM8K problems from the test set show that the curriculum-based approach yields substantial monotonic improvements in accuracy as the training horizon increases. At $h$=2 the instruct model achieves 35.06%, which increases to only 39.42% with RL on standard GSM8K problems but jumps to 56.22% when training up to a horizon of 2 and 58.51% when trained up to a horizon of 5. Similarly, at $h$=3 the instruct model achieves 20.07%, which lifts to 37.76% with a curriculum up to $h$=3. For longer horizons (harder problems), the effect of curriculum is even more visible, increasing accuracy by about $3\times$ at $h$=4 (6.70% → 18.77%) and $h$=5 (3.57% → 9.82%). We present these improvements in Figure 2.

In Table 1, the **Only-L1** baseline improves $h$=1 but shows no gains on longer horizons. Similarly, **Uniform-Mix** even at an equal training compute baseline shows no gains. **Only-Long** also leads to no long-horizon improvements due to the lack of useful training signal at longer lengths (see Section 3). Furthermore, Cui et al. (2025) show that the entropy of

a policy undergoing RL training collapses quickly, causing the improvements from RL to saturate quickly. While this is true for our baselines, our curriculum training repeatedly introduces new levels of difficulty (exploration), which allows *scaling RL for up to 5× more steps* to keep improving capabilities. We leave a deeper investigation into the scaling properties of our method to future work.

Having established that our curriculum-based training imparts new, in-domain capabilities, we now investigate whether these learned skills generalize to challenging, out-of-domain benchmarks that implicitly require long-horizon reasoning. Our results (Table 2) demonstrate that the skills acquired from solving synthetically chained problems transfer remarkably well to harder problems.

> **Curriculum RL bootstraps long-horizon reasoning**
>
> Training up to horizon $h$ **extends usable learning signal** on $h + 1$ and shifts probability mass into the long-sequence tail monotonically. For example, training to $h$=3 lifts $h$=4 from 6.70% to 15.55%; training to $h$=4 lifts $h$=5 from 3.57% to 9.23%. This provides enough training signal for the next stage, allowing curriculum learning to be extremely effective. **See theoretical analysis in Appendix B.**

**Why single-step accuracy is not enough.** In Section 3, we claim that LHR depends on more than single step accuracy. Prior to RL training, the model achieves 82.79% at $h$=1. If errors were independent, we would expect 68.54% at $h$=2 and 56.75% at $h$=3 by multiplicative compounding, yet we observe 35.06% and 20.07% (Table 1). Even after 200 RL steps (**Only-L1**), despite a slight increase at $h$=1, performance drops to 39.42% at $h$=2 and 18.37% at $h$=3 rather than 69.28% and 57.67% expected under the independent error assumption. Our results show that simply

*Table 2.* Performance on *harder math benchmarks* and OOD *long-context benchmarks* improves with GSM8K RL curriculum training stages. Bootstrapping simple existing data can be used for scaling reinforcement learning with generalizable gains. AIME avg@32.

| | | Generalization to Significantly Harder Math Problems | | | | | Long-Context | |
| --- | --- | --- | --- | --- | --- | --- | --- | --- |
| Model/setting | MATH-500 | Symbolic P1 | Symbolic P2 | MMLU Pro | AIME 2025 | AIME 2024 | Long Bench v2 | Hash hop |
| **Instruct model** | 64.20 | 67.06 | 43.08 | 58.47 | 1.77 | 5.10 | 35.00 | 15.98 |
| *Standard RLVR on GSM8K* | | | | | | | | |
| **GSM8K RLVR** | 66.20 | 71.40 | 47.60 | 60.62 | 2.71 | 6.88 | 35.30 | 14.76 |
| *Curriculum RL on Composed GSM8K Problems* | | | | | | | | |
| **Len-2 GSM8K** | 67.00 | 72.86 | 50.80 | 59.73 | 1.25 | 7.19 | 36.20 | 16.17 |
| **Len-3 GSM8K** | 66.80 | 70.70 | 49.48 | 61.21 | 1.67 | 5.73 | 37.10 | 17.62 |
| **Len-4 GSM8K** | 68.40 | 72.22 | 51.92 | 60.91 | 2.60 | 10.00 | 36.20 | **18.98** |
| **Len-5 GSM8K** | **69.20** (+7.8%) | **73.28** (+9.3%) | **52.00** (+20.7%) | **61.21** (+4.7%) | **3.02** (+70.6%) | **10.52** (+106.3%) | **37.90** (+8.3%) | 18.73 (+17.2%) |

improving per-step accuracy $p$ (Sinha et al., 2025) may not yield compounding LHR improvements, and long-horizon specific training is needed to improve long-horizon performance (which in turn improves atomic-task reliability $p$).

**Learning new capabilities with RL.** We now discuss the second part of our claim in Section 3. LHR depends on additional *horizon-dependent capabilities* $\sigma_j$ that can be improved using RL training over a curriculum. Yue et al. (2025) show an important result that RLVR on LLMs only improves the sample efficiency of reasoning capabilities already present in the base model, and no new capabilities are learnt. They observe that at a high pass@k (such as 128), capabilities of these RL trained models originate from and are bounded by the base model (with the pass@k performance quickly converging). Therefore, when an RL trained model is not bounded by the base/instruct model at high pass@k, one can empirically demonstrate that new skills are learnt.

> **LHR Training can teach new capabilities**
>
> We demonstrate for the first time that **Curriculum RL can teach new capabilities that go significantly beyond the base model even at** *pass@128*. Our curriculum based training on compositional synthetic data is therefore crucial.

Our *explicit-horizon* training and testing setting allows us to isolate out capability improvements that can go beyond the base model with only RL. Importantly, proving one of the central claims in our paper, we evaluate our final model on unseen longer horizons ($h = 6$, 7, and 8) up to a very high sampling budget *(pass@128)*. Our results in Figure 3 show that while RL on standard GSM8K is bounded by the instruct model's capabilities (and converges very quickly), our long horizon trained models perform significantly better even at high $k = 128$. The difference in performance between models trained on only $h = 1$ (which leads to almost no LHR improvements) and curriculum

trained models validates the separation between $p$ and $\sigma_j$ discussed in Section 3. This shows our method unlocks correct reasoning paths that were previously inaccessible to the model, providing genuinely new LHR capabilities $\sigma_j$. Compared to common RLVR training paradigms studied in Yue et al. (2025), we are therefore able to show that our RL method can indeed teach new reasoning skills.

In this section, we show significant gains on explicit-horizon in-domain tasks and that our model learns new reasoning capabilities with our curriculum based training. Our explicit-horizon GSM8K setting, while very useful in allowing us to isolate these capabilities and compare all training methods, is still relatively artificial. In Section 5, we therefore test our GSM8K trained LHR models on significantly harder (unseen) problems that are implicitly longer.

## 5. Generalization to harder benchmarks

**Transfer to Olympiad level math.** In Table 2, we evaluate our GSM8K long horizon trained models on MATH-500, GSM-Symbolic P1, GSM-Symbolic P2, MMLU Pro Math, and AIME. These tasks can be categorized as *implicit-horizon* and benefit significantly from LHR training on much easier *explicit-horizon* tasks. For instance, performance on GSM-Symbolic P1 goes from $67.06 \rightarrow 73.28$, P2: $43.08 \rightarrow 52.00$, and strikingly AIME 2024 from $5.10 \rightarrow 10.52$, a $2.06\times$ increase. These improvements show a transfer of the capabilities targeted in Section 4. LHR enables bootstraping capabilities from significantly easier tasks to gains on much harder ones without any extra labels or supervision. We see a scaling trend, where continued RL training on longer *explicit-horizons* leads to improvements on *harder implicit-horizon* tasks. Bootstrapping composed LHR data can allow more RL compute to be spend on the same dataset.

**Transfer to long-context benchmarks.** We now evaluate our GSM8K LHR models on OOD long-context bench-

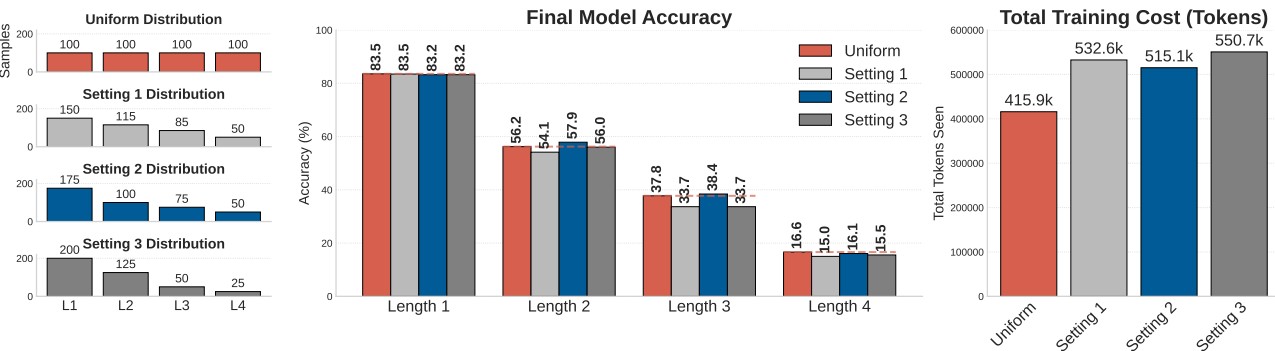

*Figure 4.* **Left**: Sample count distributions for four settings. **Middle**: Comparing accuracy at each stage across sample count settings. Under mild skew towards shorter samples like Setting 1 and 2, the model can perform as well as the uniform sample baseline. **Right**: Comparing the training compute across settings. The settings skewed towards shorter samples have more training cost in terms of training tokens seen. Overall, low-cost data distributions can achieve near-optimal performance.

marks to see if the state tracking capabilities ($\sigma_j$) from Section 3 improve. We test two main long-context benchmarks: LongBench-v2 (Bai et al., 2025) and Hash-hop (Magic, 2024). LongBench-v2 measures understanding and reasoning over QA documents, long-dialogue, repositories, etc. (8k–2M words). Hash-hop tests ultra-long-context retrieval and variable tracing by making models follow shuffled chains of random hash → hash pairs. Tables 2 and 7 summarize our results, with a $35.00\% \rightarrow 37.90\%$ improvement on LongBench-v2 and a $15.98\% \rightarrow 18.73\%$ improvement on Hash-hop, both unrelated to GSM8K.

*Table 3.* Long-horizon training on composed GSM8K problems generalizes remarkably well to OOD *ReasoningGym* domains, outperforming length-1 (standard) RLVR and the Instruct model. We use default ReasoningGym configurations for our evaluations.

| Model / setting | Generalization to ReasoningGym domains | | | |
|---|---|---|---|---|
| | Propositional logic | Graphs (largest island) | Algorithmic (sentence reorder) | Algorithmic (matrix) |
| **Instruct** | 22.90 | 15.00 | 9.60 | 2.70 |
| **RLVR** | 12.40 | 17.00 | 9.80 | 3.90 |
| **h1 (ours)** | **47.10** | **22.50** | **18.80** | **4.20** |

**Transfer to non-mathematical reasoning benchmarks.** We test long-horizon trained models on ReasoningGym (Stojanovski et al., 2025) domains to evaluate whether the *horizon-dependent reliability* improvements generalize to non-mathematical but verifiable reasoning tasks. ReasoningGym consists of a diverse set of reasoning environments allowing us to evaluate cross-domain transfer and skill generalization. Specifically, we test across logic (propositional logic), graphs (largest island), algorithmic problems (sentence reordering and matrix manipulation), arithmetic (decimal arithmetic), and geometry. These problems require working memory, graph traversal, multi-step rule follow-

ing, and correct final answers. On ReasoningGym, long-horizon training on composed GSM8K significantly outperforms both the Instruct model and RLVR trained on normal GSM8K. $h1$ generalizes from $22.90\% \rightarrow 47.10\%$ on propositional logic, $15.00\% \rightarrow 22.50\%$ on graph problems (largest island), $9.60\% \rightarrow 18.80\%$ on algorithmic sentence reordering, and $2.70\% \rightarrow 4.20\%$ on algorithmic matrix manipulation. Performance on games (game of life) goes from $76.20\% \rightarrow 74.90\%$. Overall, skills learnt from long-horizon training generalize well to out-of-distribution reasoning problems. See Table 3.

This transfer patterns aligns with our *pass@k* capability improvement results from Section 4 and our theoretical framing. Tasks requiring sequential dependent reasoning, such as AIME or GSM-Symbolic problems, benefit from improved long-horizon reasoning capabilities that were learned on much simpler composed tasks. Crucially, improvements in aspects such as state-tracking are also observable from our long-context evals. Our results indicate that a curriculum of simple explicit-horizon tasks can bootstrap advanced reasoning **(see our theoretical analysis in Appendix B)**, providing a scalable path where composing problems at the edge of what can be solved would push capabilities further without new annotations. Importantly, our method *replicates on other model families* (using the **Llama-3.2-3B Instruct** model) and additional results are presented in Appendix D.

## 6. Designing a cost-efficient curriculum

In most real-world scenarios, there is an abundance of short-horizon data, and long-horizon data is expensive to obtain (Kwa et al., 2025; Sinha et al., 2025). In this section, we ask whether long-horizon performance can be obtained from training data distributions that are "cheaper" than a uniform one. Namely, whether we can train on more short data and

less long data and still achieve the same performance. We also evaluate how this affects the required training compute.

Our experiments follow the same curriculum RL method described in Sections 3 and 4. During training, we train up to saturation for each stage, spending as much training compute as needed until there are no further improvements in accuracy. We track the total number of tokens seen by the model during training. We create four different curricula with the same total number of samples, and different proportions of short- and long-horizon data (Figure 4, Left).

The results in Figure 4 (Middle, Right) show that high long-horizon performance can be achieved even in data-constrained scenarios with training data distributions skewed towards shorter examples, but the trade-off is that more training compute needs to be spent overall. Understanding this trade-off is crucial for creating cost-effective datasets geared towards long-horizon reasoning tasks. Therefore, to further study this, we simplify our experimental setup to a SFT setting on a simpler task (multiplication), and scale up the search space for comprehensive evaluations. In Appendix C we show that, for a target accuracy, a similar trade-off exists between (i) training cost and (ii) training compute budget. As long as more training compute is spent (discussed in Appendix C), high performance can be achieved with limited long-horizon data.

## 7. Discussion

In this paper, we introduce a new method for improving long-horizon reasoning in large language models. Our method leverages existing short-horizon data by composing new, multi-step problems through a chaining process. This approach allows us to scale reinforcement learning training via a curriculum, yielding substantial performance gains on multi-step reasoning tasks. An important result of our work is that the skills learned through this curriculum transfer to new, challenging reasoning and long-context tasks. Furthermore, our results show that the trained model learns genuinely new reasoning capabilities, rather than just refining existing ones. We also demonstrate that comparable performance can be achieved even when there is abundant short-horizon data but limited long-horizon data, providing a scalable and data-efficient path for improving models.

While the goal of our paper is to introduce an early method for improving long-horizon reasoning, we see two promising directions for extensions. One is incorporating new sources of atomic skills beyond GSM8K. The other is using DAGs that expand the serial dependency structure in our current method. We believe these two paths would offer useful extensions to the framework we introduce in this paper and further improve long-horizon reasoning capabilities.

## Impact Statement

This paper presents work whose goal is to advance the field of Machine Learning. It serves to study how model performance can be improved on longer output sequences by bootstrapping from existing data.

## Acknowledgments

We would like to thank Shashwat Goel, Dimitris Papailiopoulos, Andrew Zhao, Harkirat Behl, Ameya Prabhu, Vibhav Vineet, Manan Tomar, Akshit Sinha, Dulhan Jayalath, Satwik Bhattamishra, Andis Draguns, Bhavya Kailkhura, Brian Bartoldson, Aleks Petrov, Ronald Clark, Fabio Pizzati, Vaibhav Balloli, Weishi Yan, and Alex Pondaven for their time and insightful discussions.

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

# A. Compositions and Computational Graphs.

As discussed in Section 3, we compose atomic tasks into chains of length $h$ to construct *explicit-horizon* problems. An atomic task, defined in natural language, takes an input $x_j$ and has a corresponding ground-truth answer $y_j$. These are individual GSM8K problems in our case. Let $y_j$ be the answer to sub-problem $j$. A lightweight *adapter* $\phi_j$ transforms $y_j$ into parameters for the next sub-problem:

$$y_j = f_j(x_j), \qquad x_{j+1} = \phi_j(y_j), \qquad j = 1, \ldots, h-1.$$

The final answer is then arrived at by serially composing sub-problems and adapters, such that

$$y_h = f_h(\phi_h(\cdots \phi_1(f_1(x_1)))).$$

We start from *atomic tasks* $f_j$, which we consider to be short, self-contained problems (e.g., single GSM8K questions) with a uniquely verifiable answer that the base model solves with non-trivial accuracy. An atomic task, defined in natural language, takes an input $x_j$ and has a corresponding ground-truth answer $y_j$. To create long-horizon examples, we build chains of $h$ atomic tasks so that later sub-problems depend on earlier results.

**Two practical ways we form dependencies.**

1. **Fixed-template (transformation) chaining.** We keep a fixed bank of atomic problems with fixed parameters and fixed ground-truth answers. To introduce dependency, we rewrite a parameter *symbolically* in terms of a previous answer so that its numeric value is unchanged. Thus the downstream question text now references an earlier result (asking for some transformation on it), but its correct answer remains the same as in the original problem.

2. **Substitution (recompute) chaining.** We substitute a function of a previous answer directly into the next problem's parameter(s) and *recompute* that problem's ground-truth. Here both the instance and its correct answer change. We anticipate that verifiable problems in domains such as chemistry, physics, computer science, etc. would need substitution chaining (rule based or with an LLM) for meaningful compositions.

We use *substitution chaining* for composing GSM8K problems in Section 4 and *transformation chaining* for composing MATH problems in Appendix D.

Adapters can be identity or simple deterministic transforms (e.g., scaling, unit conversion, affine transformation). Each chain of length $h$ is rendered as a single prompt listing the $h$ sub-problems in order. The model is instructed to solve them sequentially but is supervised only on the final answer $y_h$ (outcome-based verifiable rewards for reinforcement learning).[*]

We can now analyze our method from the perspective of computational graphs (Zhou et al., 2025a). Each verifiable problem, such as in GSM8K dataset, forms a single-sink directional acyclic graph where each node represents an operator consuming the value from previous nodes and producing the value for the next node(s) or as an output of the graph as shown in Figure 5. The height of the graph then can represent the number of steps that must be planned and accurately carried out (executed) while the width of the graph represents the state that must be maintained at each step and accurately manipulated. This framing can extend to the composed problems themselves. The sequential composition method presented in our paper forms a simple yet efficient technique that enables models to learn through a curriculum consisting of problems that consist of an increasingly large number of steps. We do note that the definition of what counts as a step is currently vague and dependent on the problem itself. *Future work will investigate using more general DAGs to bootstrap an RL curriculum and concretely define what counts as a step for the sake of long-horizon tasks.*

We generated the computational graphs for all of the problems in GSM8k and AIME24 datasets to examine the patterns of computation and compositions for these problems. While the graphs for these two datasets are not equivalent as they use different operators, they give some insights into why our method is able to show generalization across different datasets. For GSM8k problems, we found the computational graphs have average width and height of 4.1 and 4.0 respectively, while AIME graphs have average width and height of 6.6 and 7.1 respectively. We then compare the statistics for the graphs of AIME problems solved before and after our procedure as shown in Table 4. We observe that our method enables models to solve significantly deeper computational graphs, which we speculate generalizes to harder problems such as AIME.

---

[*]We apply standard well-posedness filters: type checks, numeric range clipping, and de-duplication.

Albert is wondering how much pizza he can eat in one day. He buys 2 large pizzas and 2 small pizzas. A large pizza has 16 slices and a small pizza has 8 slices. If he eats it all, how many pieces does he eat that day?

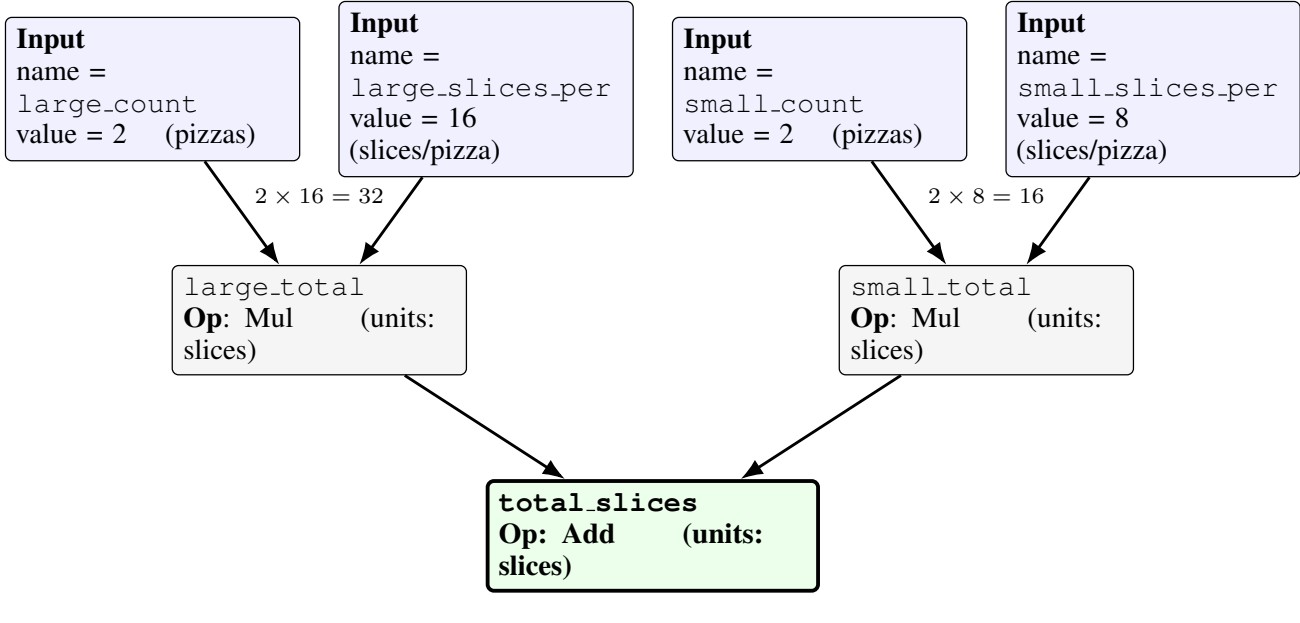

$$32 + 16 = \textbf{48}$$

*Figure 5.* Question and computational graph for a GSM8K problem (final answer: **48**).

## B. Theoretical Analysis

Intuitively, under our long-horizon skill model in Section 3, attempting to train directly on long-horizon data with outcome rewards results in vanishing gradient signal, as very few rollouts achieve any reward. Curriculum training overcomes this by initially training at short horizons, where this signal is stronger. Raising the success rate in achieving a reward at horizon $j$ also raises the success rate for horizons $> j$, and so when we come to train at $j + 1$, the signal is no longer vanishing. In the analysis below, we prove that this is the case in our simplified long-horizon skills model, and demonstrate an exponential decrease (in the horizon length $H$) in the sample complexity for curriculum training vs direct outcome reward-only horizon $H$ training, along with an equivalence between curriculum training and training at horizon $H$ with dense, per-step rewards.

### B.1. Setup and notation

We study the simplified model of long-horizon skills from Section 3 to analyse curriculum benefits for LHR. Consider a finite-horizon episodic problem with maximum horizon $H$. At each step $j \in \{1, \ldots, H\}$ there is a binary correctness indicator $C_j \in \{0, 1\}$, where $C_j = 1$ iff step $j$ is correct. Write $C_{<j} := \prod_{\ell < j} C_\ell$ and $C_{\leq j} := \prod_{\ell \leq j} C_\ell$. The policy $\pi_\theta$ (language model) generates a response $y = (y_1, \ldots, y_H)$ autoregressively by

$$y_k \sim \pi_\theta(\cdot \mid x, y_{\leq k}), \qquad k = 1, \ldots, H,$$

and we suppress explicit dependence on $x$ below. A verifier $R_i(y) \in \{0, 1\}$ pays 1 iff $y_{\leq i}$ is correct, i.e. $R_i(y) = \mathbb{I}\{C_{\leq i} = 1\}$.

*Table 4.* AIME24 solved problems comparison. N+E denotes Nodes + Edges.

|  | N+E | Width | Height |
|---|---|---|---|
| Baseline: Instruct-model solved problems | 47.25 | 8 | 6.75 |
| Ours: Additional newly solved problems | 54.3 | 4.7 | 10.3 |

As in Section 3, let $p(\theta_0) \in (0, 1]$ denote *atomic task reliability* and let $\sigma_j(\theta_j) \in [0, 1]$ denote *context-length–dependent reliability* with $\sigma_1 \equiv 1$. We assume disjoint depth parameters $\theta_j$ for $j \geq 1$ and a shared atomic block $\theta_0$, reflecting that training strictly below depth $j$ cannot improve the skill captured by $\sigma_j$. Define

$$q_j := p(\theta_0)\,\sigma_j(\theta_j) \;=\; \mathbb{P}(C_j = 1 \mid C_{<j} = 1), \qquad q_j \in (0, 1),$$

and

$$s_i \;:=\; \prod_{j=1}^{i} q_j \;=\; \mathbb{P}(C_{\leq i} = 1).$$

We assume that at initialization $q_j \in [\delta, 1 - \delta]$ for some constant $\delta \in (0, 1/2)$.

For a chosen training horizon $h$, the objective is

$$J_h(\theta) := \mathbb{E}_{y \sim \pi_\theta}[R_h(y)] \;=\; s_h,$$

and for $h = H$ we write $J(\theta) = J_H(\theta)$.

We use REINFORCE (Williams, 1992) with an unbiased advantage estimator with *constant* baseline equal to the same-horizon mean $\mu_h = s_h$ (the form approximated by GRPO (Shao et al., 2024)). For a parameter block $\theta_k$,

$$\nabla_{\theta_k} J_h(\theta) \;=\; \mathbb{E}\big[(R_h - \mu_h)\,\nabla_{\theta_k} \log \pi_\theta(y)\big], \qquad \bar{g}_k \;=\; \frac{1}{N} \sum_{i=1}^{N} (R_h(y_i) - \mu_h)\,\nabla_{\theta_k} \log \pi_\theta(y_i).$$

Because $\mu_h$ is constant, $\mathbb{E}[\bar{g}_k] = \nabla_{\theta_k} J_h(\theta)$.

## B.2. Per-update improvement and batch size

We follow (Zhang et al., 2025b) in analysing the expected improvement of our policy when taking updates $\theta_k^+ = \theta_k + \eta \bar{g}_k$. Under the assumption that $J_h$ is $L$-smooth in $\theta_k$, taking this update (holding all $\theta_{j \neq k}$ fixed) results in expected improvement (cf. Duchi, 2018; Bubeck, 2015)

$$\mathbb{E}\big[J_h(\theta^+) - J_h(\theta)\big] \;\geq\; \eta \,\|\nabla_{\theta_k} J_h(\theta)\|^2 \Big(1 - \tfrac{L\eta}{2}\Big(1 + \tfrac{1}{\mathrm{SNR}(\theta_k)}\Big)\Big), \tag{1}$$

where

$$\mathrm{SNR}_h(\theta_k) \;:=\; \frac{\|\mathbb{E}\bar{g}_k\|^2}{\mathbb{E}\|\bar{g}_k - \mathbb{E}\bar{g}_k\|^2}.$$

**Proof.** Without loss of generality, view $J_h$ as a function of $\theta_k$ only. Given $J_h$ is $L$-smooth in $\theta_k$, applying the descent lemma to $-J_h$ gives, for any $u$,

$$J_h(\theta_k + u) \;\geq\; J_h(\theta_k) + \langle \nabla_{\theta_k} J_h(\theta), u \rangle - \tfrac{L}{2}\|u\|^2.$$

With $u = \eta \bar{g}_k$, and taking expectation over the randomness in $\bar{g}_k$, we obtain

$$\mathbb{E}\big[J_h(\theta^+) - J_h(\theta)\big] \;\geq\; \eta \big\langle \nabla_{\theta_k} J_h(\theta), \mathbb{E}\bar{g}_k \big\rangle \;-\; \tfrac{L\eta^2}{2}\,\mathbb{E}\|\bar{g}_k\|^2.$$

Decomposing $\mathbb{E}\|\bar{g}_k\|^2 = \mathbb{E}\|\bar{g}_k - \mathbb{E}\bar{g}_k\|^2 \;+\; \|\mathbb{E}\bar{g}_k\|^2$ and recalling that $\mathbb{E}[\bar{g}_k] = \nabla_{\theta_k} J_h(\theta)$ yields

$$\mathbb{E}\big[J_h(\theta^+) - J_h(\theta)\big] \;\geq\; \eta\,\|\nabla_{\theta_k} J_h(\theta)\|^2 \;-\; \tfrac{L\eta^2}{2}\Big(\mathbb{E}\|\bar{g}_k - \mathbb{E}\bar{g}_k\|^2 + \|\nabla_{\theta_k} J_h(\theta)\|^2\Big),$$

and simple rearrangement results in the form in equation 1. □

Optimising over $\eta$ yields

$$\mathbb{E}\big[J_h(\theta^+) - J_h(\theta)\big] \;\geq\; \underbrace{\frac{\|\nabla_{\theta_k} J_h(\theta)\|^2}{2L}}_{:=\Delta_{k,h}^{(0)}} \cdot \frac{1}{1 + 1/\mathrm{SNR}(\theta_k)}\,. \tag{2}$$

To achieve a fraction $\beta \in (0,1)$ of the noiseless gain $\Delta_{k,h}^{(0)}$ it suffices that

$$\mathrm{SNR}_h(\theta_k) \;\geq\; \frac{\beta}{1-\beta}. \tag{3}$$

Therefore, the quantities $\Delta_{k,h}^{(0)}$ and $\mathrm{SNR}_h(\theta_k)$ determine how much we can improve our policy by, and how many samples $N$ will be required to do so.

### B.3. Signal-to-noise ratio

Fix a training horizon $h \geq k$. Abbreviate

$$s := s_{k-1}, \qquad q := q_k, \qquad T := \prod_{j=k+1}^{h} q_j,$$

so $s_h = s\,q\,T$ and $T = 1$ if $h = k$. Because $\theta_k$ only affects $q_k$,

$$\mathbb{E}[\bar{g}_k] \;=\; \nabla_{\theta_k} s_h \;=\; s\,T\,\nabla_{\theta_k} q_k.$$

For the variance, note that

$$\nabla_{\theta_k} \log \pi_\theta(C_k \mid C_{<k}{=}1) = \Big(\frac{C_k}{q} - \frac{1-C_k}{1-q}\Big) \nabla_{\theta_k} q_k = \frac{C_k - q}{q(1-q)}\,\nabla_{\theta_k} q_k,$$

and $\nabla_{\theta_k} \log \pi_\theta(y)$ vanishes unless $C_{<k} = 1$. A direct enumeration over $(C_{<k}, C_k, C_{k+1:h})$ with the constant baseline $\mu_h = s\,q\,T$ gives the conditional second moment

$$\mathbb{E}\big[(R_h - \mu_h)^2\, \|\nabla_{\theta_k} \log \pi_\theta(y)\|^2 \;\big|\; C_{<k} = 1\big] \;=\; T\,\|\nabla_{\theta_k} q_k\|^2 \left[\frac{(1-q) + s\,q\,T\,(s\,q - 2(1-q))}{q(1-q)}\right]. \tag{4}$$

We therefore obtain

$$\mathbb{E}\big\|\bar{g}_k - \mathbb{E}\bar{g}_k\big\|^2 \;=\; \frac{s}{N}\,T\,\|\nabla_{\theta_k} q_k\|^2 \left[\frac{(1-q) + s\,q\,T\,(s\,q - 3(1-q))}{q(1-q)}\right]. \tag{5}$$

The exact SNR is then

$$\mathrm{SNR}_h(\theta_k) \;=\; \frac{N\,s\,T\,q(1-q)}{(1-q) + s\,q\,T\,(s\,q - 3(1-q))}. \tag{6}$$

In the long-horizon regime $T \ll 1$, the denominator is $(1-q)\,(1 + O(T))$, so

$$\mathrm{SNR}_h(\theta_k) \;=\; N\,s\,q\,T\Big(1 + O\big(\tfrac{T}{1-q}\big)\Big) \;=\; \Theta(N\,s\,q\,T) \;=\; \Theta(N\,s_h). \tag{7}$$

### B.4. Regimes and consequences

**Single-step only (train only at $h = 1$).** When training only at $i = 1$, the required batch size and per-update noiseless improvement clearly do not depend on $H$. Training at $h = 1$ changes $p(\theta_0)$ but cannot change $\sigma_j$ for $j \geq 2$, so even $p \to 1$ leaves the long-horizon ceiling at $\prod_{j=2}^{H} \sigma_j$. Nevertheless, for any target success probability $c \in (0,1)$, improving $p$ increases the largest $h$ with $s_h \geq c$; if $\sigma_j$ are fixed, then

$$h \;=\; \frac{\ln c - \sum_{j=2}^{h} \ln \sigma_j}{\ln p}\,.$$

**Direct full horizon (train only at $h = H$).** As $q_j \in [\delta, 1 - \delta]$ at initialisation, $s_H = \Theta(\alpha^H)$ for some $\alpha \in (0, 1)$, and by equation 7

$$\mathrm{SNR}_H(\theta_k) \;=\; \Theta(N\, s_H) \;=\; \Theta(N\, \alpha^H) \qquad (k \geq 1).$$

To keep a constant fraction of the noiseless gain one must therefore take

$$N \;=\; \Theta(\alpha^{-H}),$$

while the noiseless improvement for $\theta_k$ is

$$\Delta^{(0)}_{k,H} \;=\; \frac{\|\nabla_{\theta_k} J_H\|^2}{2L} \;=\; \frac{(s_{k-1} T_{k+1:H})^2}{2L} \|\nabla_{\theta_k} q_k\|^2 \;=\; \Theta\!\big(\alpha^{2H}\, \|\nabla_{\theta_k} q_k\|^2\big).$$

The required batch size to achieve a constant fraction of the noiseless improvement is exponential, while the noiseless improvement decays exponentially, making training directly at large $H$ effectively impossible. In fact, direct training at horizon $H$ is worse than single-step training, as the signal is too small to effectively raise $p$ $\big(\mathrm{SNR}_H(\theta_0) = \Theta(N\, H\, \alpha^H)\big)$.

**Curriculum over depths (train at $h = k$ and advance when $q_k \geq 1 - \varepsilon$).** Given a target success probability $s_H \geq c \in (0, 1)$, we can ensure that curriculum training achieves this by only progressing the horizon $k$ when $q_k \geq 1 - \varepsilon$, such that $(1 - \varepsilon)^H \geq c$, and therefore $\epsilon \sim (-\ln c)/H$. If we assume that the earlier stages have been learned so $s_{k-1} \geq (1 - \varepsilon)^{k-1} \geq c$, then from equation 6 we have that,

$$\mathrm{SNR}_k(\theta_k) = \Theta\!\left(N \frac{q_k(1 - q_k)}{(2q_k - 1)^2}\right).$$

The SNR is minimised near the target $q_k = 1 - \varepsilon$, where $\frac{q_k(1-q_k)}{(2q_k-1)^2} \asymp \varepsilon$. To realise a fixed fraction of $\Delta^{(0)}_{k,k}$ it suffices, by equation 3, to take

$$N \;=\; \Theta\!\left(\frac{1}{\varepsilon}\right) \;=\; \Theta(H).$$

At $h = k$, $\nabla_{\theta_k} J_k = s_{k-1} \nabla_{\theta_k} q_k$, so the noiseless gain is

$$\Delta^{(0)}_{k,k} \;=\; \frac{\|\nabla_{\theta_k} J_k\|^2}{2L} \;=\; \frac{s_{k-1}^2}{2L} \|\nabla_{\theta_k} q_k\|^2 \;=\; \Theta\big(\|\nabla_{\theta_k} q_k\|^2\big),$$

independent of $H$. Provided $\|\nabla_{\theta_k} q_k\|^2$ does not decay faster than polynomially as $q_k \to 1 - \varepsilon$, curriculum yields polynomial sample complexity in $H$.

**Uniform mixture over lengths.** Sample $I \sim \mathrm{Unif}\{1:H\}$ and run the depth-$I$ estimator; for parameters $\theta_i$, the per-*iteration* SNR obtained for its update averages to

$$\mathbb{E}_I[\mathrm{SNR}_I(\theta_i)] \;=\; \Theta\!\left(\frac{N}{H}\, s_{i-1}\, q_i \sum_{t=0}^{H-i} T_{i+1:i+t}\right).$$

*Frontier phase.* We say horizon $i$ is at the frontier when earlier skills are sufficiently reliable while deeper ones are not yet learnt, namely

$$s_{i-1} \geq c \quad \text{for some fixed } c \in (0,1) \qquad \text{and} \qquad \sum_{t=0}^{H-i} T_{i+1:i+t} = \Theta(1).$$

During this frontier phase,

$$\mathbb{E}_I[\mathrm{SNR}_I(\theta_i)] \;=\; \Theta\!\left(\frac{N}{H}\, s_{i-1}\, q_i\right).$$

Whenever we sample a batch with $I = i$, we obtain the same noiseless improvement and batch size scaling as curriculum training, with

$$N = \Theta\left(\frac{1}{\epsilon}\right) = \Theta(H), \qquad \Delta_{i,i}^{(0)} = \Theta\left(\|\nabla_{\theta_i} q_i\|^2\right).$$

Whenever we sample $I \neq i$, we see negligible change as samples with $h < i$ cannot improve $q_i$, and samples with $h > i$ have per-iteration gain that scales with $s_{h-1}^3$.

$$\mathbb{E}_I[\Delta_h \text{ per iter}] = \Theta\left(\frac{N}{H} s_{h-1}^3 \|\nabla_{\theta_i} q_h\|^2\right).$$

Therefore, it takes $\sim H$ times longer to train with uniform sampling than with curriculum, due to only a fraction $1/H$ of the updates being "useful" at each frontier $i \in \{1, \cdots, H\}$.

## B.5. Dense rewards

We now analyse the reward-to-go estimator (Williams, 1992; Sutton et al., 1999) with *dense rewards*. For a fixed training horizon $h \geq k$, define

$$R_t(y) := \mathbb{I}\{C_{\leq t} = 1\}, \qquad G_k(y) := \sum_{t=k}^{h} R_t(y),$$

so $G_k$ is the return-to-go from step $k$. As before, write

$$s := s_{k-1} = \prod_{j<k} q_j, \qquad q := q_k, \qquad T_{k+1:k+t} := \prod_{j=k+1}^{k+t} q_j.$$

On *reach* ($C_{<k} = 1$), $C_k \sim \text{Bernoulli}(q)$ and the tail process after $k$ yields

$$Q_k^* := 1 + \sum_{t=1}^{h-k} T_{k+1:k+t}, \qquad \Sigma_k := 1 + \sum_{t=1}^{h-k} (2t+1) T_{k+1:k+t}, \qquad V_k := \Sigma_k - (Q_k^*)^2,$$

so that, given $C_k = 1$, $\mathbb{E}[G_k \mid C_{<k} = 1, C_k = 1] = Q_k^*$ and $\text{Var}(G_k \mid C_{<k} = 1, C_k = 1) = V_k$. The *unconditional* mean return-to-go at step $k$ is

$$b_k := \mathbb{E}[G_k] = \sum_{t=k}^{h} s_t = s\, q\, Q_k^*. \tag{8}$$

We use the unbiased reward-to-go REINFORCE block with the constant baseline $b_k$:

$$g_k = (G_k - b_k)\, \nabla_{\theta_k} \log \pi_\theta(y), \qquad \bar{g}_k = \frac{1}{N} \sum_{i=1}^{N} g_k^{(i)}.$$

Since only step $k$ depends on $\theta_k$ and $C_{<k} = 1$ is required for a contribution,

$$\nabla_{\theta_k} \log \pi_\theta(y) = \nabla_{\theta_k} \log \pi_\theta(C_k \mid C_{<k}=1) = \left(\frac{C_k}{q} - \frac{1 - C_k}{1 - q}\right) \nabla_{\theta_k} q_k = \frac{C_k - q}{q(1 - q)} \nabla_{\theta_k} q_k.$$

**Mean.** Using $\mathbb{E}[(G_k - b_k) \mid C_{<k} = 1] = \mathbb{E}[G_k \mid C_{<k} = 1] - b_k = q\, Q_k^* - b_k$ and $\mathbb{E}[C_k - q \mid C_{<k} = 1] = 0$, the cross term vanishes and

$$\mathbb{E}\bar{g}_k = \mathbb{E}g_k = \mathbb{E}\left[G_k \nabla_{\theta_k} \log \pi_\theta(y)\right] = s\, \frac{\nabla_{\theta_k} q_k}{q(1 - q)} \mathbb{E}[C_k(C_k - q)\, G_k \mid C_{<k}=1]$$

$$= s\, \frac{\nabla_{\theta_k} q_k}{q(1 - q)} (1 - q)\, q\, Q_k^* = s\, Q_k^* \nabla_{\theta_k} q_k. \tag{9}$$

**Second moment and variance.** Condition on reach ($C_{<k}=1$). Writing $C := C_k$ and $Y$ for the tail variable so that $G_k = CY$ on reach,

$$\|g_k\|^2 = \frac{\|\nabla_{\theta_k} q_k\|^2}{q^2(1-q)^2} (C-q)^2 (CY - b_k)^2.$$

Taking expectation over $C \in \{0,1\}$ and using $\mathbb{E}\big[(Y-b_k)^2 \mid C=1\big] = V_k + (Q_k^* - b_k)^2$, we obtain the conditional second moment

$$\mathbb{E}\big[\|g_k\|^2 \mid C_{<k}=1\big] = \|\nabla_{\theta_k} q_k\|^2 \left[\frac{b_k^2}{1-q} + \frac{V_k + (Q_k^* - b_k)^2}{q}\right]. \tag{10}$$

Unconditioning multiplies by $s$ (probability of reach), and substituting $b_k = s\, q\, Q_k^*$ from equation 8 yields

$$\mathbb{E}\|g_k\|^2 = s\, \|\nabla_{\theta_k} q_k\|^2 \left[\frac{V_k}{q} + (Q_k^*)^2\left(\frac{1}{q} - 3s + s^2 q + \frac{s^2 q^2}{1-q}\right)\right]. \tag{11}$$

Subtracting $\|\mathbb{E}g_k\|^2 = s^2 (Q_k^*)^2 \|\nabla_{\theta_k} q_k\|^2$ from equation 11 gives the unconditional variance of the single-trajectory block:

$$\mathbb{E}\|g_k - \mathbb{E}g_k\|^2 = s\, \|\nabla_{\theta_k} q_k\|^2 \left[\frac{V_k}{q} + (Q_k^*)^2\left(\frac{1}{q} - 3s + s^2 q + \frac{s^2 q^2}{1-q}\right)\right]. \tag{12}$$

For the batch average $\bar{g}_k$ this variance scales as $1/N$.

**Signal-to-noise ratio.** Combining equation 9 and equation 12 and accounting for the $1/N$ factor,

$$\mathrm{SNR}_{\mathrm{dense}}(\theta_k) = \frac{N\, s\, (Q_k^*)^2}{\dfrac{V_k}{q} + (Q_k^*)^2\left(\dfrac{1}{q} - 3s + s^2 q + \dfrac{s^2 q^2}{1-q}\right)} \tag{13}$$

$$= \frac{N\, s\, q\, (1-q)}{\dfrac{\Sigma_k}{(Q_k^*)^2}(1-q) + sq(sq - 3(1-q))}. \tag{14}$$

This form is the same as that in the terminal rewards only case, except with the ratio $(Q_k^*)^2/\Sigma_k$ replacing $T$.

**Regular tails.** Under our initialization assumption that $q_j \in [\delta, 1-\delta]$ with fixed $\delta \in (0, 1/2)$, for all $t \geq 1$, $\delta^t \leq T_{k+1:k+t} \leq (1-\delta)^t$, and so

$$1 + \delta \leq Q_k^* \leq \frac{1}{\delta}, \qquad 1 + 3\delta \leq \Sigma_k \leq 1 + \sum_{t \geq 1}(2t+1)(1-\delta)^t = 1 + \frac{(1-\delta)(2+\delta)}{\delta^2}.$$

This gives

$$\frac{(1+\delta)^2}{1 + \frac{(1-\delta)(2+\delta)}{\delta^2}} \leq \frac{(Q_k^*)^2}{\Sigma_k} \leq \frac{1/\delta^2}{1+3\delta} = \Theta(1).$$

Therefore, before the tail has been trained, the SNR for $\theta_k$ scales as

$$\mathrm{SNR}_{\mathrm{dense}}(\theta_k) = \frac{N\, s\, q\, (1-q)}{(1-q) + sq(sq - 3(1-q))},$$

the same as in the curriculum case.

**Training.** We should see similar *frontier* behaviour as we see in the uniform mixture case, as the SNR scales with the success probability $s_i$. Again, we say horizon $i$ is at the frontier if $s_{i-1} \geq c$ for some fixed $c \in (0, 1)$. To make $q_i \geq 1 - \varepsilon$ (so that $s_i \geq c$) we require

$$N \; = \; \Theta\left(\frac{1}{\varepsilon}\right) \; = \; \Theta(H),$$

while the noiseless improvement is

$$\Delta^{(0)}_{k,\text{dense}} \; = \; \Theta\big(\|\nabla_{\theta_k} q_k\|^2\big).$$

Hence, we have the same scaling and sample complexity as in curriculum learning. The wall clock time may be slightly lower as we do not prevent training at longer lengths $\geq i$ before $s_{i-1} \geq 1 - \varepsilon$.

> **Key Insight**
>
> With terminal-only rewards, the signal-to-noise ratio scales with the success rate, making direct full-horizon training exponentially inefficient in the horizon length. Curriculum learning or dense rewards solve this by progressively raising the success rate at increasing horizons, reducing the sample complexity required to make meaningful policy improvements from exponential to polynomial.

## C. Decreasing Sample Complexity for Longer Training Data

In Section 6, we train with the sample distributions mentioned in Table 5 up to saturation. From the explicit-horizon accuracies on our test set, we can see that the performance obtained from a uniform distribution of $n$-horizon data can be matched with a more cost efficient skewed distribution as long as more compute is spent. However, this is a small search space that primarily shows such a property could exist. Here, we provide a more rigorous set of experiments to support this finding.

*Table 5.* Performance trade-offs for training with skewed sample distributions. By increasing the training budget, performance on most length bins can be recovered (**Uniform distribution** vs **Settings 1, 2, and 3**) even with a cost-efficient dataset skewed heavily towards shorter examples.

| Setting | Sample Distribution (L1/L2/L3/L4) | Training Steps (L1/L2/L3/L4) | Accuracy per Length Bin (%) | | | | Training Tokens |
|---|---|---|---|---|---|---|---|
| | | | L1 | L2 | L3 | L4 | |
| *Baseline: Uniform Data Distribution* | | | | | | | |
| **Baseline** | 100 / 100 / 100 / 100 | 100 / 110 / 120 / 150 | 83.55 | 56.22 | 37.75 | 16.62 | 415 940.00 |
| *Skewed Data Distributions (Constant Sample Count)* | | | | | | | |
| **Setting 1** | 150 / 115 / 85 / 50 | 150 / 180 / 250 / 100 | 83.47 | 54.14 | 33.67 | 15.01 | 532 610.00 |
| **Setting 2** | 175 / 100 / 75 / 50 | 175 / 220 / 240 / 70 | 83.24 | 57.88 | 38.43 | 16.08 | 515 110.00 |
| **Setting 3** | 200 / 125 / 50 / 25 | 200 / 230 / 220 / 100 | 83.24 | 56.02 | 33.67 | 15.54 | 550 730.00 |

To support Section 6 better, we scale up the search space by first simplifying our experimental setting. In particular, we consider training a 135M-parameter GPT-2 model on integer multiplication problems through SFT. We generate the multiplication problems by sampling two operands, and writing out the chain of computations. We define **length** as the sum of number of digits of both operands, analogous to the number of chained GSM8K problems in our primary setting. Then we can separate the training dataset into bins grouped by distinct lengths. We vary the length distribution of training dataset by varying the samples in each length bins. Finally we associate a **cost** to each data length, which represents the cost of generating the data. This metric mirrors the real-world concern that longer data is harder to collect.

## C.1. Trade-off between data cost and compute

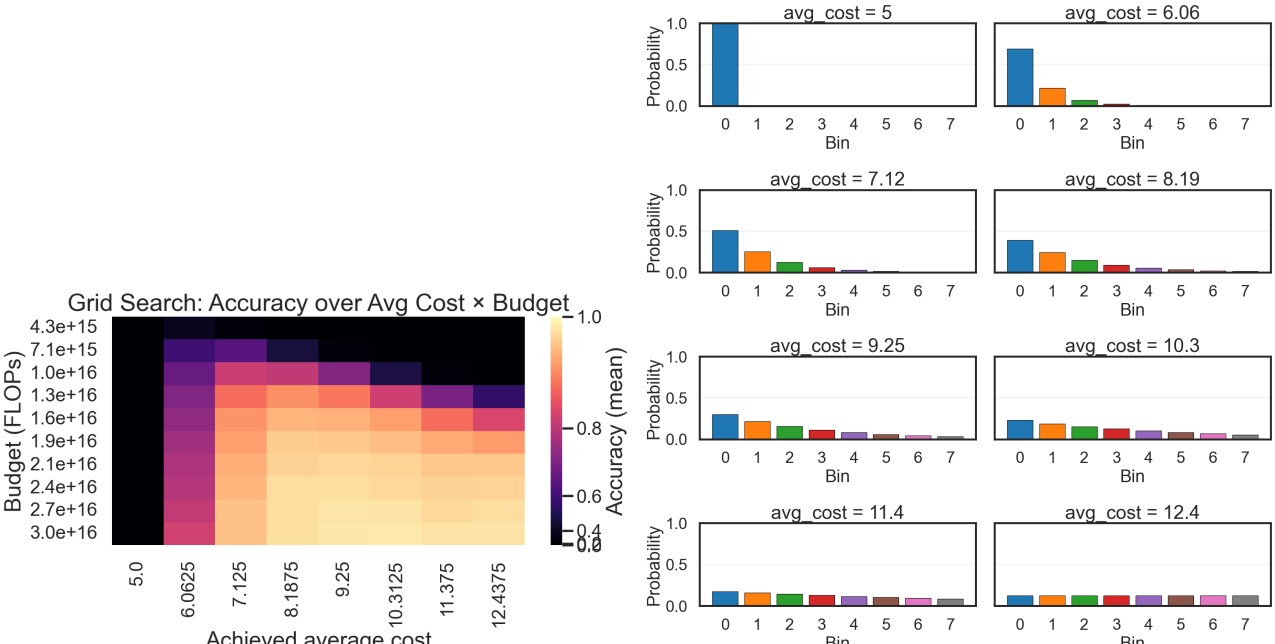

*Figure 6.* Right: We parameterize different training distributions using a single cost scalar plotted against FLOP budget. Left: Sweeping different choices of training budget and distribution costs.

Using the multiplication task, we study the trade-off between (1) skewedness towards shorter lengths in the training distribution and (2) total training budget. Figure 6 sweeps over many choices of budget and cost. A first observation is that for the same target accuracy, a training run can either have a lower cost data distribution and use more budget, or vise versa. Figure 6 also shows other relationships. For example, at each training budget there appears to be a set of optimal data distributions. To achieve the same performance with a cheaper distribution than the optimal ones requires increasing the training budget. For distributions with a lot more long samples beyond the optimal point, the rate of learning seems to decrease. We believe this is because as we shift more weight to longer examples, the multiplication becomes harder to learn overall. This motivates us to investigate what the most cost efficient (optimal) set of distributions is for a given training budget.

### C.2. Searching for the minimum cost distribution under the same budget

By principally testing different data mixtures above, we show that a curriculum can achieve optimal performance even when using limited long-horizon samples. Here, we investigate what the set of low cost distributions looks like at a fixed training budget. We create 3 length bins, which evenly divide the sample lengths in our dataset. We keep a fixed FLOP budget and vary the data mixture based on the length bins to find the distributions with the least average cost that still achieve high performance after training. We visualize this search procedure in Figure 7, which shows a distinct feasible region for the 3-bin probability distribution where training runs are successful.

## D. Additional Experimental Details.

In Section 4, we trained the Qwen-2.5-3B Instruct model (Qwen et al., 2025) using the hyperparameters outlined in Table 6. Training was conducted for 200 optimization steps for each horizon length in our curriculum, where each step processed a single sample, for a total of 200 samples per horizon. We utilized the Dr. GRPO algorithm (Liu et al., 2025c), which builds on top of the Group-Relative Policy Optimization (GRPO) training objective (Shao et al., 2024) by removing the length and variance normalization terms. For each problem length, we evaluated the model every 50 steps and selected the best checkpoint based on validation performance. This checkpoint was then used as the initialization for training on the subsequent, longer-horizon problem length.

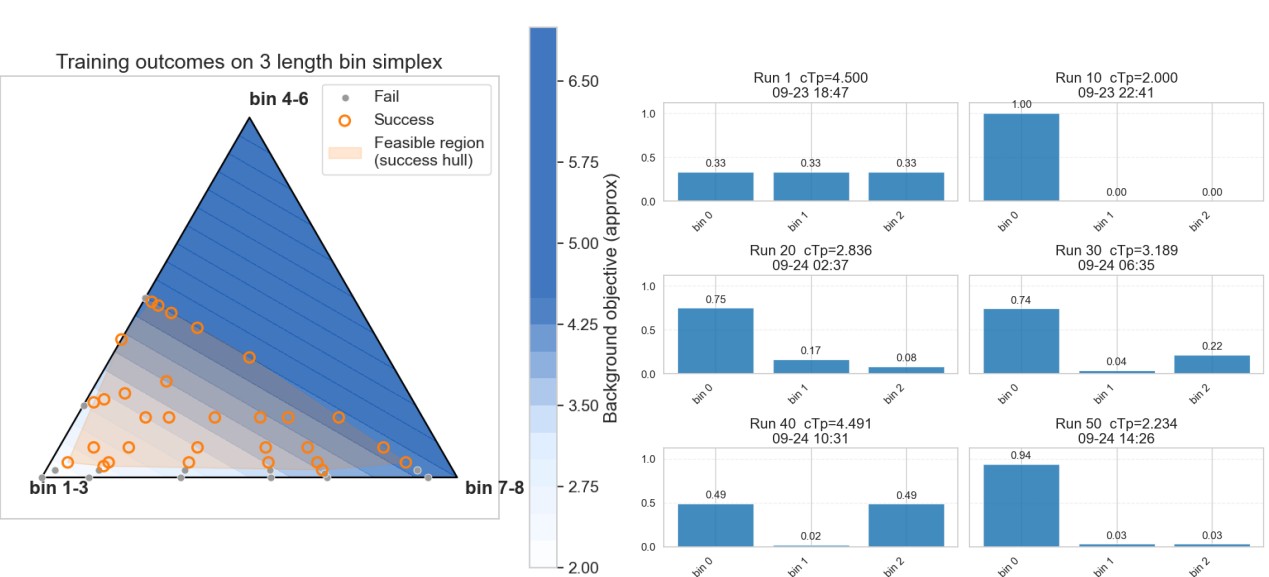

*Figure 7.* Left: Training trails with 3 length bin data distribution, plotted over the probability simplex. The blue gradient is the *cost* of the distribution, as defined in C.1. Each point on the simplex is a training run with the specific data distribution. We start from the uniform distribution (middle point) and send 8 rays to the cheaper half of the simplex boundary (we did not explore the more expensive half). Then we bisect each ray to find the feasibility boundary along it. Overall, there is a convex feasible region that forms close to the simplex boundaries, and we are able to find cost efficient data distributions. Right: Examples of different 3-bin training distributions during the search.

The max completion length was dynamically adjusted based on the number of sub-problems to accommodate the increasing reasoning horizon. Specifically, we used a maximum completion length of 768 for 1-subproblem tasks, 1024 for 2-subproblem tasks, 1280 for 3-subproblem tasks, and 1536 for both 4- and 5-subproblem tasks. Dynamically increasing the completion length allows the model enough token space to solve the problem while also constraining it to the maximum length required to complete the task. We implemented GRPO training using the Verifiers library (Brown, 2025).

---

**System prompt for RL training**

Respond in the following format, with only the numerical answer between the <answer> tags:
<reasoning>
. . .
</reasoning>
<answer>
. . .
</answer>

---

The same hyperparameters were applied to our baselines in Table 1. For these baselines, we trained up to an equal amount of compute as our main experiments and selected the best checkpoint from each run based on validation performance.

To evaluate our model's generalization capabilities, we performed a series of zero-shot evaluations (sampling temperature 0.1) on a variety of benchmarks, including AIME 2024, AIME 2025, MMLU Pro Math (Wang et al., 2024), GSM Symbolic (Mirzadeh et al., 2025), MATH-500 (Hendrycks et al., 2021), LongBench-v2(Bai et al., 2025), Hash-hop(Magic, 2024), and GPQA(Rein et al., 2024) (Tables 2, 7). For the Hash-hop benchmark, we computed the average accuracy across multiple settings, including context lengths of 10k, 20k, and 30k characters, and 1, 2, 3, and 4 hops.

Besides harder math benchmarks, our curriculum-based training generalizes to benchmarks that require long-context and complex reasoning, even though our models were only trained on composed mathematical problems. The results on LongBench-v2, Hash-hop, and GPQA show a consistent improvement in performance as the training horizon increases, demonstrating that our method imparts transferable skills such as state-tracking and the ability to reason over long

*Table 6.* Hyperparameters used for the curriculum-based RL training stages across different models.

| Parameter | Qwen 2.5 3B Instruct | Qwen 2.5 7B Instruct | Llama 3.2 3B Instruct |
|---|---|---|---|
| Training Steps per Horizon | 200 | 200 | 200 |
| Samples per Horizon | 200 | 200 | 200 |
| Number of Generations per Prompt | 16 | 16 | 16 |
| Learning Rate | $5 \times 10^{-6}$ | $2 \times 10^{-6}$ | $5 \times 10^{-6}$ |
| Learning Rate Scheduler | Cosine | Cosine | Cosine |
| Warmup Steps | 30 | 30 | 20 |
| Max Gradient Norm | 0.1 | 0.1 | 0.1 |
| Loss Type | Dr. GRPO | Dr. GRPO | Dr. GRPO |

*Table 7.* Performance on long context benchmarks improves significantly with GSM8K RL curriculum training stages. Training on increasing complexity of GSM8K leads to strong out-of-domain generalization with better state tracking and context management capabilities.

| | Generalization to Long Context Benchmarks | | |
|---|---|---|---|
| **Model / setting** | **LongBench-v2** | **Hash-hop** | **GPQA** |
| **Instruct model** | 35.00 | 15.98 | 25.00 |
| *Standard RLVR on GSM8K* | | | |
| **GSM8K RLVR** | 35.30 | 14.76 | 26.56 |
| *Curriculum RL on Composed GSM8K Problems* | | | |
| **Len-2 GSM8K** | 36.20 | 16.17 | 25.22 |
| **Len-3 GSM8K** | 37.10 | 17.62 | 26.12 |
| **Len-4 GSM8K** | 36.20 | **18.98** | 26.34 |
| **Len-5 GSM8K** | **37.90** 
 (+8.3%) | 18.73 
 (+17.4%) | **27.23** 
 (+8.9%) |

sequences. For example, performance on LongBench-v2 increases from 35.00% (untrained) to 37.90% after training up to a 5-subproblem horizon. These results are presented in Table 7.

In order to strengthen our results and demonstrate the generality of our method, we apply it to two additional models: Qwen-2.5-7B Instruct and Llama-3.2-3B Instruct. The training hyperparameters for all models are detailed in Table 6.

For the Qwen-2.5-7B Instruct model, we construct chained problems from a more challenging source, the MATH dataset. We connect subproblems with integer-valued answers by applying simple operations (e.g., addition or subtraction) to generate the numerical input for the next problem. Our training shows a strong, consistent performance lift, with mean accuracy on multi-step problems rising from 45.50% to 50.65% (Table 8). This improvement transfers to out-of-domain benchmarks, validating the method's ability to generalize beyond the specific training domain.

*Table 8.* Long Horizon MATH Training on **Qwen-2.5-7B Instruct**. Curriculum stages lead to improvements in in-domain performance and generalization metrics. We believe these results will improve further with better hyperparameter tuning and more training compute. Currently, this result aims to show that our method can indeed transfer to larger models and harder training datasets.

| Model / setting | Accuracy on MATH Problems | | | | | Accuracy on Harder Problems | |
|---|---|---|---|---|---|---|---|
| | Len-1 | Len-2 | Len-3 | Len-4 | Mean | Symbolic P2 | LongBench v2 |
| **Instruct model** | 74.00 | 52.60 | 29.40 | 26.00 | 45.50 | 61.36 | 33.60 |
| *Standard RLVR on MATH* | | | | | | | |
| **MATH RLVR** | 76.20 | 53.80 | 32.00 | 29.00 | 47.75 | 64.96 | 34.50 |
| *Curriculum RL on Composed MATH Problems* | | | | | | | |
| **Len-2 MATH** | **77.00** | 56.20 | 34.40 | 27.66 | 48.82 | **65.60** | 34.50 |
| **Len-3 MATH** | 76.20 | 56.00 | 35.40 | 28.86 | 49.12 | 65.32 | 34.50 |
| **Len-4 MATH** | 76.80 | **56.60** | **37.80** | **31.40** | **50.65** | 64.88 | **35.30** |
| | (+4.1%) | (+7.6%) | (+28.6%) | (+20.8%) | (+11.3%) | (+6.9%) | (+5.1%) |

To show the applicability of our method to another model family, we use the Llama-3.2-3B Instruct model with the same chained GSM8K problems as our primary experiments in Section 4. The results for this model are presented in Table 9. We show that our method can be successfully applied to a different model family, confirming that our curriculum learning based RL training on composed synthetic data can lead to robust and generalized long-horizon reasoning capabilities.

*Table 9.* Long Horizon GSM8K Training on **Llama-3.2-3B Instruct**. Curriculum RL leads to significant improvements in in-domain performance and general long-horizon capabilities.

| Model / setting | Accuracy on GSM8K Problems | | | | | Accuracy on Harder Problems | |
|---|---|---|---|---|---|---|---|
| | Len-1 | Len-2 | Len-3 | Len-4 | Mean | Symbolic P1 | AIME Mean |
| **Instruct model** | 78.00 | 11.83 | 4.42 | 1.34 | 23.90 | 54.84 | 2.09 |
| *Curriculum RL on Composed GSM8K Problems* | | | | | | | |
| **Len-1 GSM8K** | 79.00 | 13.28 | 7.14 | 2.14 | 25.39 | 55.16 | 2.87 |
| **Len-2 GSM8K** | 80.20 | 35.06 | 15.99 | 6.43 | 34.42 | 57.52 | 2.92 |
| **Len-3 GSM8K** | **80.60** | **35.27** | **17.35** | **6.70** | **34.98** | **57.75** | **3.18** |
| | (+3.3%) | (+198.1%) | (+292.5%) | (+400.0%) | (+46.4%) | (+5.3%) | (+52.2%) |

# E. Qualitative Example

> **LHR Trained Qwen-2.5-3B on Explicit-Horizon Problems**
>
> **Long-Horizon Question**
> Solve the following math problem step by step:
> Step 1: Janet's ducks lay 16 eggs per day. She eats three for breakfast every morning and bakes muffins for her friends every day with four. She sells the remainder at the farmers' market daily for $2 per fresh duck egg. How much in dollars does she make every day at the farmers' market?
> Step 2: Take your final answer from Step 1 and substitute it for W in the following problem:
> Ella earned $W for 3 hours of dog walking. If she continues to walk dogs at the same rate, how much would she earn after 12 hours?
> Write out the updated version of the problem with the number from Step 1 in place of W.
> Step 3: Solve the updated problem from Step 2 step by step.
> Step 4: Take your final answer from Step 3 and substitute it for U in the following problem:
> Peter purchased U popsicles at $0.25 each. He also purchased 4 ice cream bars at $0.50 each. How much did he pay in total in dollars?

Write out the updated version of the problem with the number from Step 3 in place of U.
Step 5: Solve the updated problem from Step 4 step by step.
Step 6: Take your final answer from Step 5 and substitute it for Y in the following problem:
Fred was preparing for a party to be held in four days. So, he made Y gallons of root beer on the first day and put them in the refrigerator cooler. But later that evening, his children discovered the delicious nectar and robbed the cooler, drinking 4 of those gallons of root beer. On the second day, his wife Barbie also discovered the root beer and accidentally spilled 7 gallons. On the third day, Fred's friend Ronnie visited Fred's house and helped himself to the root beer, further reducing the amount remaining by 5 gallons. On the fourth day, 3 people showed up for the party. If Fred and the others shared the remaining root beer equally, how much was available for each to drink during the party?
Write out the updated version of the problem with the number from Step 5 in place of Y.
Step 7: Solve the updated problem from Step 6 step by step.
Step 8: Take your final answer from Step 7 and substitute it for Z in the following problem:
Craig has 2 twenty dollar bills. He buys six squirt guns for $2 each. He also buys Z packs of water balloons for $3 each. How much money does he have left?
Write out the updated version of the problem with the number from Step 7 in place of Z.
Step 9: Solve the updated problem from Step 8 step by step. In the end, provide only the final numerical answer.

**Ground truth answer:** 25

**Untrained model response**

<reasoning>

Janet's ducks lay 16 eggs per day. She eats 3 eggs for breakfast and uses 4 eggs to make muffins. The remaining eggs are sold at the farmers' market. The number of eggs sold is calculated as follows:

$$16 - 3 - 4 = 9$$

Since she sells each egg for $2, her daily earnings at the farmers' market are:

$$9 \times 2 = 18$$

So, Janet makes $18 every day at the farmers' market. This value is substituted for W in the next problem.

Ella earned $18 for 3 hours of dog walking. If she continues to walk dogs at the same rate, her hourly rate is:

$$\frac{18}{3} = 6$$

Ella's hourly rate is $6. This value is substituted for W in the updated problem.

Ella earns $6 per hour. If she continues to walk dogs at the same rate, how much would she earn after 12 hours?

$$6 \times 12 = 72$$

So, Ella would earn $72 after 12 hours. This value is substituted for U in the next problem.

Peter purchased $72 at $0.25 each. He also purchased 4 ice cream bars at $0.50 each. The total cost is calculated as follows:
$$72 \times 0.25 + 4 \times 0.50 = 18 + 2 = 20$$

So, Peter paid $20 in total. This value is substituted for U in the next problem.

Peter paid $20. If he purchased 20 packs of root beer, how much was available for each to drink during the party?

$$\frac{20}{4} = 5$$

So, 5 gallons of root beer were available for each to drink during the party. This value is substituted for Y in the next problem.

Fred made 5 gallons of root beer. On the second day, his wife spilled 7 gallons, on the third day, his friend Ronnie helped himself to 5 gallons, and on the fourth day, 3 people showed up for the party. The remaining root beer is calculated as follows:
$$5 - 7 - 5 - 3 = -10$$

Since the remaining amount is negative, it means Fred ran out of root beer and did not have any left for the party. This value is substituted for Y in the next problem.

Fred did not have any root beer left. If he purchased 0 gallons of root beer, how much money does he have left after buying 2 twenty dollar bills?
$$2 \times 20 = 40$$

So, Fred spent $40 on the 2 twenty dollar bills. This value is substituted for Y in the next problem.

Craig has $40. He buys 6 squirt guns for $2 each and 3 packs of water balloons for $3 each. The total cost is calculated as follows:
$$6 \times 2 + 3 \times 3 = 12 + 9 = 21$$

So, Craig spent $21 on the squirt guns and water balloons. The amount of money he has left is:

$$40 - 21 = 19$$

So, Craig has $19 left. This value is substituted for Z in the next problem.

Craig has $19 left. If he buys Z packs of water balloons for $3 each, how much money does he have left?

$$19 - 3 \times 3 = 19 - 9 = 10$$

So, Craig has $10 left. This value is substituted for Z in the next problem.

Craig has $10 left. If he buys 10 packs of water balloons for $3 each, how much money does he have left?

$$10 - 10 \times 3 = 10 - 30 = -20$$

Since the remaining amount is negative, it means Craig does not have enough money to buy 10 packs of water balloons. The final numerical answer is -20.
</reasoning>
<answer>
-20
</answer>

**Analysis (Errors in the untrained model's response)**
**State-tracking errors**

1. **Step 6→7 (after "Peter paid $20"):** Took shortcut $20/4 = 5$ and then *reused* $Y = 5$ as the starting gallons for Fred ("Fred made 5 gallons..."). Correct carry-over for Step 6 is $Y = 20$.

2. **Step 8→9 (Craig stage):** From "Craig has $19 left." it set $Z = 19$ but computed $19 - 3 \times 3$ instead of $19 - 3Z$. Next line used $Z = 10$, showing inconsistent substitution.

**Problem-logic errors**

1. **Step 6 / start of Step 7:** Treated $20 as gallons and skipped events, doing $20/4$ instead of $20 - 4 - 7 - 5 = 4$ then $4/4 = 1$.

2. **Step 7:** Subtracted people as gallons, computing $5 - 7 - 5 - 3$ instead of dividing the remaining gallons among 4 people.

**LHR model response**

<reasoning>

1. First, we need to calculate how many eggs Janet sells each day.
   - Janet's ducks lay 16 eggs per day.
   - She eats 3 eggs for breakfast.
   - She bakes muffins with 4 eggs.
   So, the number of eggs she sells each day is:

$$16 - 3 - 4 = 9 \text{ eggs}$$

Janet sells the eggs at \$2 per egg. Therefore, the amount she makes every day at the farmers' market is:

$$9 \times 2 = \$18$$

2. Substitute the final answer from Step 1 (18) into the problem:
   Ella earned \$18 for 3 hours of dog walking. If she continues to walk dogs at the same rate, how much would she earn after 12 hours?

3. Ella's rate of earning is \$18 for 3 hours. Therefore, her hourly rate is:

$$\frac{18}{3} = \$6 \text{ per hour}$$

To find out how much she would earn after 12 hours, we calculate:

$$6 \times 12 = \$72$$

4. Substitute the final answer from Step 3 (72) into the problem:
   Peter purchased 72 popsicles at \$0.25 each. He also purchased 4 ice cream bars at \$0.50 each. How much did he pay in total in dollars?

5. The cost of the popsicles is:
$$72 \times 0.25 = \$18$$

The cost of the ice cream bars is:
$$4 \times 0.50 = \$2$$

The total cost is:
$$18 + 2 = \$20$$

6. Substitute the final answer from Step 5 (20) into the problem:
   Fred was preparing for a party to be held in four days. So, he made 20 gallons of root beer on the first day and put them in the refrigerator cooler. But later that evening, his children discovered the delicious nectar and robbed the cooler, drinking 4 of those gallons of root beer. On the second day, his wife Barbie also discovered the root beer and accidentally spilled 7 gallons. On the third day, Fred's friend Ronnie visited Fred's house and helped himself to the root beer, further reducing the amount remaining by 5 gallons. On the fourth day, 3 people showed up for the party. If Fred and the others shared the remaining root beer equally, how much was available for each to drink during the party?

7. The amount of root beer remaining after each event is:

$$20 - 4 - 7 - 5 = 4 \text{ gallons}$$

Fred and 3 others (including himself) share the remaining 4 gallons equally. Therefore, each person gets:

$$\frac{4}{4} = 1 \text{ gallon}$$

8. Substitute the final answer from Step 7 (1) into the problem:
Craig has 2 twenty dollar bills. He buys six squirt guns for $2 each. He also buys Z packs of water balloons for $3 each. How much money does he have left?

9. The cost of the squirt guns is:
$$6 \times 2 = \$12$$

The cost of the water balloons is:
$$Z \times 3$$

The total cost is:
$$12 + 3Z$$

Craig has 2 * 20 = $40. After spending, the remaining money is:

$$40 - (12 + 3Z)$$

Substitute the final answer from Step 7 (1) for Z:

$$40 - (12 + 3 \times 1) = 40 - 15 = \$25$$

Final numerical answer: 25

</reasoning>
<answer>
25
</answer>

