# OpenReview forum: "h1: Bootstrapping LLMs to Reason over Longer Horizons via Reinforcement Learning"
_ICML.cc/2026/Conference — ICML 2026 spotlight_

### Official Review · Reviewer_2jzn · 2026-02-18

**Soundness:** 3
**Presentation:** 3
**Significance:** 3
**Originality:** 3
**Overall Recommendation:** 5
**Confidence:** 3

**Summary:**

The paper introduces h1, an RL-based method to improve LLMs' long-horizon reasoning capabilities. The method works by composing short-horizon reasoning problems to synthesize long-horizon reasoning problems and using a curriculum-based RL framework that progressively trains the LLMs by increasing the reasoning complexity. Experiments on challenging benchmarks show that models trained on composed GSM8K data can achieve significantly improved performance on more challenging benchmarks.

**Compliance With Llm Reviewing Policy:**

Affirmed.

**Final Justification:**

Question resolved.

**Key Questions For Authors:**

please refer to weaknesses

**Limitations:**

yes

**Strengths And Weaknesses:**

Strengths:
1. The paper is well-written and easy to follow
2. To my knowledge, synthetic data for long-horizon reasoning is novel, and the motivation is clear and intuitive.
3. Comprehensive experiments on various benchmarks, and 2 models of different families are used to demonstrate that the method is generalizable.
4. Well-articulated ablation and analysis.

Weaknesses/Questions:
1. The claim that 'new skills' are learnt under h1 is reasonable but not definitive; a more in-depth discussion of this may be beneficial.
2. Since h1 is explicitly centered around long-horizon reasoning, I'm curious if this will cause the LLMs to produce overly and unnecessarily long reasoning steps for simpler questions, i.e. will it significantly raise inference token cost?

---

> ### Author Rebuttal · Authors · 2026-03-29
>
> We thank the reviewer for their time and valuable feedback. We appreciate the opportunity to address the points raised and further clarify our work.
>
> 1. We refer to new skills in the context of [1]. The exact phrasing used in [1] is: “average pass@k across a dataset thus reflects the proportion of problems a model can potentially solve within k trials, offering a more robust view of its reasoning boundary. This provides a rigorous test on whether the RLVR training yields fundamentally transcending capacity, enabling the model to solve problems that the base model cannot.” New skills are defined as coverage over a finite problem set and whether pass@k performance exceeds the base model’s boundary and not necessarily as the emergence of identifiably novel strategies or algorithms. We will therefore discuss this with more nuance as the reviewer suggests and clearly mention what we mean by new skills learnt (along with a qualitative study of reasoning traces) in the camera-ready version.
>
> 2. The reviewer raises a good point about the token usage of our long-horizon trained models on simpler questions. We see largely no increase in token usage on a set of shorter horizon reasoning problems and just an increase in accuracy, which will therefore not raise inference costs much more. We use Dr. GRPO as our RL algorithm which fixes the length bias issue in GRPO. We will discuss this more clearly and also add the token usage table in our paper. These results are provided below:
>
>     Average Token Usage
>
>    | Trained on \ Tested on | L1 | L2 | L3 | L4 | L5 |
>    |--|--:|--:|--:|--:|--:|
>    | **Instruct** | 287 | 414 | 560 | 737 | 871 |
>    | **L1** | 238 | 375 | 512 | 669 | 865 |
>    | **L2** | 315 | 445 | 620 | 820 | 1100 |
>    | **L3** | 331 | 453 | 643 | 827 | 1052 |
>    | **L4** | 302 | 460 | 655 | 826 | 1036 |
>    | **L5** | 331 | 446 | 633 | 830 | 1014 |
>
>    Overall, L5 training does not increase the response length that we see when training at L2 and above. There is a slight increase from Instruct/L1 but the accuracy improvements are significantly higher to make up for this small trade-off.
>
>    This shows that our method does not cause long-horizon reasoning models to produce unnecessarily long reasoning outputs on shorter problems, demonstrating its efficiency while increasing accuracy.
>
> We sincerely appreciate the reviewer’s feedback and thank them for the opportunity to clarify our paper further. We would be very grateful if the reviewer would consider raising their score further.
>
> [1] Does Reinforcement Learning Really Incentivize Reasoning Capacity in LLMs Beyond the Base Model?, Yue et. al. NeurIPS 2025 Oral
>
> [2] Understanding R1-Zero-Like Training: A Critical Perspective, Liu et. al. COLM 2025

---

> > ### Author Rebuttal · Reviewer_2jzn · 2026-04-02
> >
> > My concerns are largely resolved and I appreciate that the author agrees to include the clarifications in the camera-ready version. I have revised my score accordingly.

---

### Official Review · Reviewer_doag · 2026-03-06

**Soundness:** 4
**Presentation:** 4
**Significance:** 4
**Originality:** 4
**Overall Recommendation:** 6
**Confidence:** 4

**Summary:**

This paper introduces h1, a method that leverages existing short-horizon data to improve long-horizon reasoning ability. By chaining multiple short-horizon problems to become an interdependent, long-horizon task, the authors show that this creates a curriculum that enables the model to continuously improve without saturating quickly. Results show great improvements on both in-domain and out-of-domain tasks, and even introduce new patterns in RL as evidenced by the rise of pass@k.

**Compliance With Llm Reviewing Policy:**

Affirmed.

**Final Justification:**

My comments are fully resolved. I raised the score to 6 and I would love to see this work published in ICML. Good work!

**Key Questions For Authors:**

Please see the weaknesses.

**Limitations:**

yes

**Strengths And Weaknesses:**

Strengths:
- The idea of chaining abundant short-horizon data to improve long-horizon reasoning is clever, and the empirical results shows strong improvements on in-domain accuracy, out-of-domain generalization, and even pass@k, which are all significant.
- Importantly, it is not easy to scale up training data for long-horizon learning. Therefore, it is a sound contribution to show that it is possible to leverage existing short-horizon data to achieve gains in a long-horizon regime.
- Pass@k improvements are also significant. In the active debate of whether RL post-training is only sharpening the distribution, this work provides a strong and meaningful counterargument.
- The idea and implementation are quite simple, which makes it easy to reproduce while providing strong results.
- The paper is very well organized and easy to follow. The problem statement and the method are also well formulated, and I like the insights into why horizons are hard in lines 183-213.


Weaknesses:
- How the authors actually perform chaining remains a bit unclear. Is it an automatic process? Do the authors use another LLM or manually perform chaining? In Appendix B, the authors mentioned: "We substitute a function of a previous answer directly into the next problem’s parameter(s) and recompute that problem’s ground-truth" which seems hard to be an automatic process, or at least requires a capable-enough LLM to do this.
- In Section 3, the authors also mentioned: "We apply basic well-posedness checks (type consistency, unit compatibility, de-duplication)" which I have the same question as above, i.e., are these done by an LLM or manual verification.
- Some analysis and statistics on the valid/invalid cases identified from the well-posedness checks would also be helpful. Overall, I'm not sure how difficult the whole chaining process is and how much it costs.
- I found it less informative to show relative gains like +180% in Table 1. I would suggest reporting absolute gain for better clarity.
- Overall, I don't have strong reasons to reject this paper, but would appreciate the answers to the above questions. I'm willing to raise my score even further if the questions are well addressed.

---

> ### Author Rebuttal · Authors · 2026-03-29
>
> We thank the reviewer for their time and valuable feedback. We appreciate the opportunity to further clarify our work.
>
> 1. Our general problem chaining method for mathematics is an automatic process. We can use regex to extract the numerical values from questions in the training set and use a simple transformation function (addition, subtraction, multiplication, division) to link numerical values across subproblems in order to compose them. This process is rule-based and we provide a straightforward example with two questions below:
>
>     **Q1:** Weng earns $12 an hour for babysitting. Yesterday, she babysat for 50 minutes. How much did she earn?
>     **A1:** $10
>
>     **Q2:** Betty is saving money for a new wallet which costs \$100 but has only half of the money needed. Her parents give her 15 dollars and her grandparents give her twice as much. How much more money does Betty need to buy the wallet?
>     **A2:** $5
>
>     To link Q2 to Q1, we extract a numerical parameter in Q2 (e.g. $100), and use a transformation function (e.g. multiplication). Then we find the answer to X * A1 = 100, giving X = 10. This leads us to the composed question:
>
>     **Q1:** Weng earns $12 an hour for babysitting. Yesterday, she babysat for 50 minutes. How much did she earn?
>     **Q2:** Betty is saving money for a new wallet which costs {10 * A1} but has only half of the money needed. Her parents give her $15 and her grandparents give her twice as much. How much more money does Betty need to buy the wallet?
>
>     **A:** $5
>
>     Substitution chaining instead requires a recomputation function per seed problem, as when parameters change, the answer must be recomputed. For datasets with step-by-step solutions like GSM8K, producing these functions is itself largely automatable: the step-by-step solutions can be extracted with regex and wrapped as parameterised Python functions, making the per-problem cost low. For other domains, such as chemistry, the recomputation functions may require more involved logic (e.g. manipulating SMILES strings or using libraries like rdkit). However, each such function, once written, allows us to generate combinatorially more data by varying the parameters and composing with arbitrary other problems. An LLM can assist in writing these functions, but the functions themselves are deterministic and human-reviewable. The additional advantage of substitution chaining is that recomputing problems with new parameters minimises benchmark contamination risks, allowing for a cleaner study. We will provide concrete examples for some other domains in our camera-ready version, to make it easier to understand the generality of composing questions as a data generation technique.
>
> 2. Our de-duplication checks are programmatic, and our type consistency/unit compatibility/answer leakage tests are simple checks that are done by an LLM.
>
> 3. We thank the reviewer for their question and the opportunity to clarify our chaining process further. We perform a clean study of the same here. We use our regex extraction and simple transformation function to develop a set of 200 composed horizon 5 problems. We then use an LLM (GPT-5 mini) to perform the checks we mention above. In only 7 cases out of the 200 are issues identified. Of these 7, 5 issues are from errors in the individual GSM8K test questions the model identifies, and 2 are because regex matching did not match every instance of an integer being matched (for e.g. the number 50 can also appear as fifty, which would not get matched). Overall, almost all automatically chained questions pass through this step, which makes our process simple and efficient.
>
> 4. We will report absolute gains instead for more clarity and thank the reviewer for suggesting this.
>
> We sincerely appreciate the reviewer’s feedback and thank them for the opportunity to clarify our paper further.

---

> > ### Author Rebuttal · Reviewer_doag · 2026-04-01
> >
> > Thank the authors for their responses, and my comments are fully resolved. I raised the score to 6 and I would love to see this work published in ICML. Good work!

---

### Official Review · Reviewer_3Mbx · 2026-03-09

**Soundness:** 3
**Presentation:** 3
**Significance:** 3
**Originality:** 3
**Overall Recommendation:** 4
**Confidence:** 3

**Summary:**

This paper proposes h1, a method for improving long-horizon reasoning in LLMs by composing short-horizon problems (primarily GSM8K) into serial dependency chains, and then training with outcome-only RL under a stage-wise curriculum over horizon length. The core claim is that long-horizon reasoning cannot be reduced to merely increasing single-step accuracy, and that curriculum RL over composed tasks teaches horizon-specific capabilities such as intermediate-state tracking and value reuse. The paper reports strong gains on synthetic explicit-horizon GSM8K chains, along with transfer to harder implicit-horizon benchmarks including MATH-500, GSM-Symbolic, AIME, LongBench-v2, Hash-hop, and several ReasoningGym domains. The method is intentionally simple: deterministic adapters map one sub-answer into the next sub-problem, no step supervision or PRMs are used, and training uses only final-answer rewards. The paper also argues that curriculum training changes the sample-complexity regime from exponential to polynomial in horizon, and includes a compute–data tradeoff discussion.

**Compliance With Llm Reviewing Policy:**

Affirmed.

**Final Justification:**

See reply to rebuttal.

**Key Questions For Authors:**

The paper is generally well written, though a few claims around “new capabilities” should be softened.

I expect to see stronger baselines with more advanced training set (e.g., MATH, DAPO-17k, etc.)

**Strengths And Weaknesses:**

S1. This work investigates a fundamental challenge in current RL-for-LLM reasoning: verifiable reward data is abundant at short horizons but scarce at long horizons, while direct full-horizon RL quickly becomes sparse and unstable. The paper frames this problem clearly and proposes a practical solution that is much more scalable than collecting dense process supervision.

S2. The method of chaining atomic tasks with lightweight deterministic adapters is elegant. The fact that the final answer alone can supervise the entire chain is the main attraction here. The paper also makes the curriculum structure operationally concrete: train stagewise on horizon 1 through horizon 5, carrying parameters forward across stages. That makes the proposal reproducible and conceptually clean.

S3. The most convincing result is not just that h1 beats the instruct model, but that it beats Only-L1, Uniform-Mix, Only-Long, and even a ToT-style inference baseline on explicit-horizon GSM8K. In Table 1, the gains at longer horizons are substantial: for example, L-4 improves from 6.70 to 18.77 and L-5 from 3.57 to 9.82 under Len-5 curriculum training. Meanwhile, Uniform-Mix collapses badly and Only-L1 barely helps the long-horizon settings. This pattern supports the paper’s claim that how long-horizon data is presented matters, not just whether it exists.

S4. Training only on composed GSM8K and still improving AIME 2024 from 5.10 to 10.52, MATH-500 from 64.20 to 69.20, LongBench-v2 from 35.00 to 37.90, Hash-hop from 15.98 to 18.73, and ReasoningGym propositional logic from 22.90 to 47.10 is a notable set of results. I especially appreciate that the paper does not restrict itself to math benchmarks; the Hash-hop and ReasoningGym results are important because they test variable tracing and non-mathematical dependency following.

W1. The strongest evidence in the paper comes from explicit-horizon GSM8K chains (Table 1, Section 4), but both training and in-domain evaluation are generated from the same atomic source family, with the same serial composition logic, and with adapters that are described as simple deterministic transforms such as scaling, addition/subtraction, and unit conversion. The evaluation therefore heavily rewards the exact capability being trained: answer extraction, answer carrying, answer transformation, and reinsertion into the next prompt.

W2. I would expect stronger baselines rather than just training models on GSM8k. To me, It sounds like the authors cherry-pick a super easy dataset that no one actually use for RL training as the main experiment.

W3. I like Figure 3 and think it is one of the strongest parts of the paper. However, I do not think it fully establishes the paper’s claim that curriculum RL “teaches genuinely new capabilities.” What it establishes more conservatively is: 1. the base/instruct model does not easily sample these successful long-horizon trajectories; 2. standard RLVR on L1 does not unlock them; 3. curriculum training on composed chains does unlock them, even at high k. It still leaves open whether the “new capability” is a fundamentally new reasoning ability.

---

> ### Author Rebuttal · Authors · 2026-03-29
>
> We thank the reviewer for their time and valuable feedback. We appreciate the opportunity to address the points raised and further clarify our work.
>
> 1. Our evaluations are on in-domain problems and several other reasoning benchmarks. In-domain evaluations naturally reward the capabilities being trained, as the test data is drawn from the same distribution as the training data. However, our stance is that the capabilities being trained (answer extraction, carrying, transformation) are important for long-horizon reasoning in general, as demonstrated by the generalisation of models trained only on chained GSM8K problems to MATH, GSM-Symbolic, MMLU Pro, AIME, LongBench-v2, Hash-hop, and several ReasoningGym problems. These questions are not simple chains (out-of-domain), and yet they experience large improvements via our training method, as the reviewer also notes.
>
> 2. Our goal is to show that long-horizon training data can be developed even on standard datasets and our method of stagewise curriculum training over this data generalizes to much harder benchmarks. For 3B parameter models, GSM8K is a widely used dataset and we provide some examples of published papers that do so [1][2][3][4][5][6]. This also fits within our compute budget, but we have begun training our primary model on the MATH training dataset (which is more compute-intensive) and will present these results in our camera-ready version, along with the additional results already in Appendix E.
>
> 3. We thank the reviewer for noting the importance of Figure 3 and giving us the opportunity to further clarify our results. We refer to new capabilities or skills in the context of [2]. The exact phrasing used in [2] is: “average pass@k across a dataset thus reflects the proportion of problems a model can potentially solve within k trials, offering a more robust view of its reasoning boundary. This provides a rigorous test on whether the RLVR training yields fundamentally transcending capacity, enabling the model to solve problems that the base model cannot.” New capabilities/skills are defined as coverage over a finite problem set and whether pass@k performance exceeds the base model’s boundary, and not necessarily as the emergence of identifiably novel strategies or algorithms. We will therefore discuss this with more nuance, as the reviewer suggests and clearly mention what we mean by new skills learnt (along with a qualitative study of reasoning traces) in the camera-ready version.
>
> We sincerely appreciate the reviewer’s feedback and have aimed to address all the comments above.
>
> [1] SimpleRL-Zoo: Investigating and Taming Zero Reinforcement Learning for Open Base Models in the Wild, Zeng et. al. COLM 2025
>
> [2] Does Reinforcement Learning Really Incentivize Reasoning Capacity in LLMs Beyond the Base Model?, Yue et. al. NeurIPS 2025 Oral
>
> [3] Representation-Based Exploration for Language Models: From Test-Time to Post-Training, Tuyls et. al. ICLR 2026
>
> [4] Echo Chamber: RL Post-training Amplifies Behaviors Learned in Pretraining, Zhao et. al. COLM 2025
>
> [5] CPPO: Accelerating the Training of Group Relative Policy Optimization-Based Reasoning Models, Lin et. al. NeurIPS 2025
>
> [6] Hybrid Latent Reasoning via Reinforcement Learning, Yue et. al. NeurIPS 2025

---

> > ### Author Rebuttal · Reviewer_3Mbx · 2026-04-02
> >
> > I thank the authors for the clear and thoughtful rebuttal. The clarification that the learned skills (answer extraction, propagation, and transformation) serve as general building blocks for long-horizon reasoning—and their demonstrated transfer to diverse benchmarks—partially addresses my concern about overfitting to the synthetic chain format. I also appreciate the justification for using GSM8K at this model scale and the commitment to include results on more challenging datasets (e.g., MATH) in the camera-ready.
> >
> > Regarding “new capabilities,” the clarified definition based on pass@k coverage is reasonable, and I agree that the claims should be framed more precisely, as the authors propose.
> >
> > While some concerns remain (particularly around disentangling general reasoning from format-specific learning), the rebuttal strengthens the paper overall. I will **maintain a positive score**.

---

### Official Review · Reviewer_C5E2 · 2026-03-11

**Soundness:** 4
**Presentation:** 2
**Significance:** 3
**Originality:** 3
**Overall Recommendation:** 4
**Confidence:** 4

**Summary:**

This paper studies the training of large language models on long-horizon reasoning tasks. The authors propose a data synthesis method that composes short-horizon problems into arbitrarily long long-horizon data with reasoning dependencies. Based on these synthesized data, they train models using stagewise curriculum RL, using outcome-only rewards. Experiments show that this method outperforms several reinforcement learning baselines. Moreover, after training on tasks with explicit horizons, the model also exhibits strong transfer to benchmarks with implicit horizons. The theoretical section further analyzes, under a simplified setting, the sample complexity advantages of curriculum-based reinforcement learning.

**Compliance With Llm Reviewing Policy:**

Affirmed.

**Key Questions For Authors:**

1. The paper repeatedly emphasizes that this data synthesis process “requires no new annotation.” In practice, however, when constructing chained problems, how are numerical values extracted from and substituted into the answers? Is this process fully rule-based, or does it introduce additional hand-crafted rules or implicit annotation costs?
2. Compared with existing stage-wise or curriculum-based reinforcement learning methods, what is the core difference and advantage of your approach? Is the main contribution the data construction strategy of synthesizing long-dependency data from short problems, or the optimization strategy of stage-wise training on fixed horizons?
3. In curriculum training, when exactly does the model move to the next stage? Is it based on a fixed number of steps, validation saturation, or some success-rate threshold? The descriptions in the main text and appendix appear slightly inconsistent, so further clarification would be helpful.

**Limitations:**

yes

**Strengths And Weaknesses:**

**Strengths:**

1. The paper provides an insightful characterization of the additional capabilities required for long-horizon reasoning. The authors decompose long-horizon reasoning performance into atomic reliability  p and horizon-dependent reliability ( $\sigma_j$ ). The latter captures the extra demands that arise as the reasoning chain becomes longer, such as maintaining intermediate states, tracking results, and composing information across steps. The experimental findings are broadly consistent with this interpretation: improving short-chain capability alone is insufficient to improve long-chain performance, whereas curriculum training substantially enhances long-horizon reasoning.
2. The paper discusses the trade-off between data distribution and training compute, analyzing whether “more short-horizon data plus less long-horizon data” can approximate the effect of a uniform distribution. This part offers potentially useful guidance for practical training resource allocation.

**Weaknesses:**

1. The paper does explain that long-chain data are constructed via chaining and deterministic adapters, and the appendix further distinguishes between transformation chaining and substitution chaining. However, the main text is still not sufficiently specific about how numerical values are extracted from the original problem/answer pairs, transformed into inputs for the next problem, and how labels are automatically recomputed. In particular, the paper lacks a clearer description of the full data generation pipeline, which limits reproducibility.
2. In the main text, curriculum learning proceeds in stages according to the horizon, but the principle for deciding when to move to the next stage is not clearly specified in the method section. The appendix states that each horizon is trained for 200 steps, with validation every 50 steps and the best checkpoint selected before advancing to the next stage; meanwhile, another experiment in Section 6 says the model is trained “up to saturation.” This suggests that the authors do have a concrete procedure in practice, but the paper lacks a unified and clear criterion for stage transition in the main text.

---

> ### Author Rebuttal · Authors · 2026-03-29
>
> We thank the reviewer for their time and valuable feedback. We appreciate the opportunity to address the points raised and further clarify our work.
>
> __Our data synthesis method:__
>
> Our general problem chaining method for mathematics is an automatic process. We use regex to extract the numerical values from questions in the training set and use a simple transformation function (addition, subtraction, multiplication, division) to link numerical values across subproblems in order to compose them. This process is rule-based, and we provide a straightforward example with two questions below:
>
> **Q1:** Weng earns $12 an hour for babysitting. Yesterday, she babysat for 50 minutes. How much did she earn?
> **A1:** $10
>
> **Q2:** Betty is saving money for a new wallet which costs \$100 but has only half of the money needed. Her parents give her 15 dollars and her grandparents give her twice as much. How much more money does Betty need to buy the wallet?
> **A2:** $5
>
> To link Q2 to Q1, we extract a numerical parameter in Q2 (e.g. $100), and use a transformation function (e.g. multiplication). Then we find the answer to X * A1 = 100, giving X = 10. This leads us to the composed question:
>
> **Q1:** Weng earns $12 an hour for babysitting. Yesterday, she babysat for 50 minutes. How much did she earn?
> **Q2:** Betty is saving money for a new wallet which costs {10 * A1} but has only half of the money needed. Her parents give her $15 and her grandparents give her twice as much. How much more money does Betty need to buy the wallet?
> **A:** $5
>
> Substitution chaining instead requires a recomputation function per seed problem, as when parameters change, the answer must be recomputed. For datasets with step-by-step solutions like GSM8K, producing these functions is itself largely automatable: the step-by-step solutions can be extracted with regex and wrapped as parameterised Python functions, making the per-problem cost low. For other domains, such as chemistry, the recomputation functions may require more involved logic (e.g. manipulating SMILES strings or using libraries like rdkit). However, each such function, once written, allows us to generate combinatorially more data by varying the parameters and composing with arbitrary other problems. An LLM can assist in writing these functions, but the functions themselves are deterministic and human-reviewable. The additional advantage of substitution chaining is that recomputing problems with new parameters minimises benchmark contamination risks, allowing for a cleaner study. We will provide concrete examples for some other domains in our camera-ready version, to make it easier to understand the generality of composing questions as a data generation technique.
>
> __Our contributions:__
>
> Our core contribution is synthesising long-horizon data from existing short-horizon data, and demonstrating that a stage-wise curriculum is essential to use this data. The optimisation strategy is a by-product of the data construction method. We then show that this leads to improvements even at high pass@k, and transfers to other benchmarks even beyond mathematics (Math-500, GSM-Symbolic P1 and P2, MMLU Pro, AIME, LongBench-v2, Hash-hop, and several ReasoningGym domains), which are all novel contributions.
>
> __Curriculum strategy:__
>
> We train up to 200 steps, after which we start seeing diminishing returns at a given horizon and wish to keep our method simple, standardized, and suitable for limited compute environments. For precision and tracking, we compute the validation accuracy every 50 steps and in almost all cases, the last checkpoint has the highest validation accuracy and is the one picked. The goal of this stagewise procedure is ease of use. In Section 6, we study a different setting where the goal is to investigate the data-compute tradeoff. For this, we train up to saturation, which means that we checkpoint every 10 steps and evaluate the point at which performance crosses some minimum threshold and when validation accuracy improvements saturate (i.e. the relative difference between the current and next 2 checkpoints is less than 3%, allowing us to compute the early stopping point for a given horizon). This allows us to study precisely how much compute is needed to achieve gains at a given horizon for our data-compute tradeoff setup. However, this is an intensive process that requires checkpointing very often and serves as a thorough data-compute tradeoff study, whereas our main method is practical and can therefore be adopted easily.
>
> We sincerely appreciate the reviewer’s feedback and have aimed to address all the comments above.

---

> > ### Author Rebuttal · Reviewer_C5E2 · 2026-04-04
> >
> > Thanks for detailed response. I have maintained score.

---

### Decision · Program_Chairs · 2026-04-30

**Decision:**

Accept (spotlight)

**Comment:**

This paper introduces h1, a framework that bootstraps long-horizon reasoning by synthetically chaining abundant short-horizon problems into complex dependency chains and training via a stage-wise curriculum RL. The reviewers were highly positive, praising the method's simplicity, reproducibility, and its ability to close the "horizon gap" using only outcome-based rewards.

A standout contribution is the characterization of reasoning performance as a combination of atomic reliability and horizon-dependent scaling. Reviewers C5E2 and 3Mbx highlighted the impressive zero-shot transfer: models trained only on composed 6th-grade math (GSM8K) achieved significant gains on competition-level benchmarks (AIME, MATH-500) and diverse domains like ReasoningGym and LongBench-v2. This suggests the model learns generalizable skills like intermediate-state tracking rather than just task-specific patterns.

During the discussion, authors successfully addressed concerns regarding the data generation pipeline and potential verbosity. They clarified that the chaining process is rule-based and automated, and demonstrated that long-horizon training does not result in unnecessarily long reasoning for simpler tasks. Notably, Reviewer doag emphasized the significance of the pass@k improvements, which provide a strong counterargument to the narrative that RL only "sharpens" existing distributions. By demonstrating both theoretical sample-complexity advantages and robust empirical generalization, this work provides a scalable path for reasoning-centric RL.